# Supercoiled DNA recognition and cleavage control in topoisomerase VI

Daniel E. Richman[1], Timothy J. Wendorff[1,6], Fahad Rashid [1], Curtis Beck[1], Qianyun Yan[1], Haley R. Johnson [2,3], Ryan A. Eckerty[3], Jonathan M. Fogg[3,4], Matthew L. Baker [5], Lynn Zechiedrich [2,3,4] & James M. Berger [1] ✉

Type II topoisomerases modulate DNA supercoiling and resolve chromosome entanglements. Type IIB topoisomerases, exemplified by DNA topoisomerase VI (Top6), are used by plants and archaea to support endoreduplication and cell proliferation, respectively; homologs of Top6 further serve to initiate meiotic recombination in eukaryotes and constitute the nuclease portion of MksBEFG/Wadjet/Gabija bacterial defense systems. To understand how such factors act upon DNA, we determine structures of Top6 bound to supercoiled minicircles in cleaved and uncleaved states using single-particle electron cryo-microscopy. The structures show that Top6 binds a curved 74 bp region of the supercoiled minicircle DNA and that it cuts at a distinct deformability motif, explaining its preference for supercoiled substrates and highlighting the role of DNA plasticity in cleavage site selection. Dynamic protein-DNA interactions and an unanticipated tension sensor help recognize bent DNA and couple ATPase disposition to cleavage state activation. Our observations explain how DNA recognition and cleavage by type II topoisomerases are regulated by interdependent structural changes in DNA and the enzyme.

Topoisomerases resolve topological problems arising from the double helical structure of DNA, controlling the supercoiling, knotting, and catenation states of chromosomes and plasmids during replication, transcription, and other DNA transactions (reviewed in McKie et al.[1]). Type II topoisomerases change the topological relationship between two DNA duplexes by transiently cleaving both strands of one double helix (the gate, or G, segment), passing the second duplex (the transport, or T, segment) through the break, and then re-sealing the break in an ATP-dependent manner[2] (Supplementary Fig. 1a). Type II topoisomerases fall into two broad categories[3,4]. Type IIA enzymes include topoisomerase II (Top2)[5,6] in eukaryotes and DNA gyrase[7] and topoisomerase IV (Top4)[8] in bacteria. Type IIB topoisomerases include archaeal and plant topoisomerase VI (Top6), which support chromosome unlinking and endoreduplication, respectively[3,9–12].

Top6 is a heterotetramer composed of two A (Top6A) and two B subunits (Top6B)[3]. Top6A houses the DNA-binding and cleavage activities of the enzyme, while Top6B serves as the site of ATP binding and hydrolysis[3,13,14] (Fig. 1a and Supplementary Fig. 1a). The structural organization of Top6 is distinct from its type IIA counterparts[15], but the two protein families share several evolutionarily conserved functional elements, including a GHKL (gyrase/Hsp90/His-kinase/MutL) ATPase module[16], a metal-ion binding topoisomerase-primase (TOPRIM) fold[17], and a winged helix (WH) DNA-binding domain that bears a catalytic tyrosine residue responsible for strand scission[13]. The TOPRIM and WH domains of Top6 reside in its A subunit, which is an ortholog of Spo11, a protein that cleaves DNA to initiate chromosomal recombination during meiosis[3,18]. More distantly related TOPRIM dimers are also employed

[1]Department of Biophysics & Biophysical Chemistry, Johns Hopkins University School of Medicine, Baltimore, MD, USA. [2]Graduate Program in Quantitative & Computational Biosciences, Baylor College of Medicine, Houston, TX, USA. [3]Department of Molecular Virology & Microbiology, Baylor College of Medicine, Houston, TX, USA. [4]Verna and Marrs McLean Department of Biochemistry & Molecular Pharmacology, Baylor College of Medicine, Houston, TX, USA. [5]Department of Biochemistry & Molecular Biology, UTHealth Houston, Houston, TX, USA. [6]Present address: Genentech, Inc., South San Francisco, CA, USA. ✉e-mail: jmberger@jhmi.edu

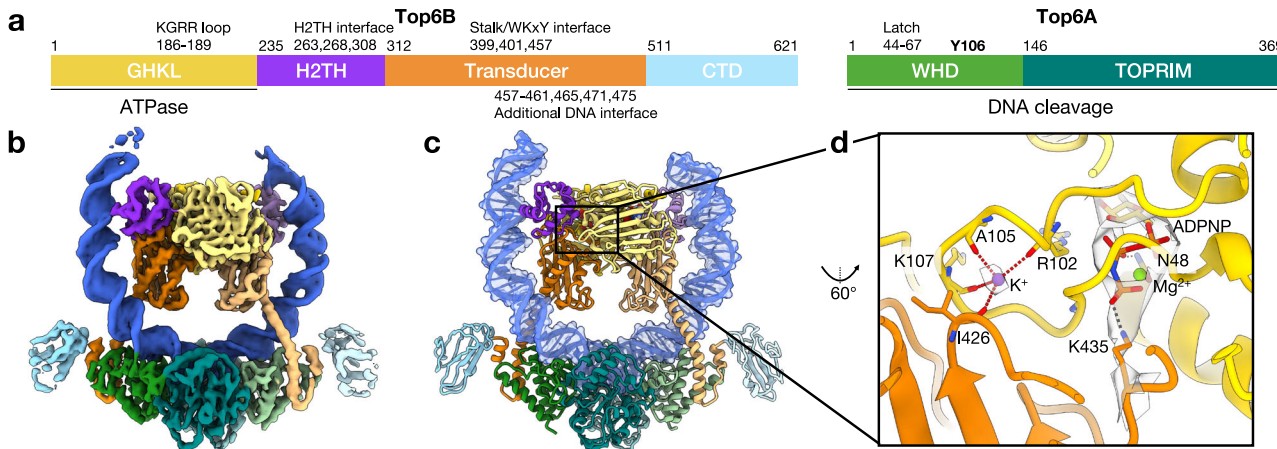

**Fig. 1 | Overview of WT Top6•mcDNA•ADPNP structure. a** Summary of the primary structure of Top6 with domains and key DNA-binding and active site locations annotated. **b** CryoEM density map (EMD-70232) with one Top6A-Top6B heterodimer colored as in (**a**) and the other in lighter versions of those colors. All Top6 figures follow this color scheme unless otherwise noted. **c** Cartoon representation of the atomic model derived from the map (PDB ID: 9O8P). The DNA is shown with a semitransparent atomic surface. **d** Detail of the ADPNP- and metal-binding region of GHKL and transducer domains in a B-subunit monomer. Chemical moieties and elements are colored conventionally, including the potassium ion (purple) and magnesium ion (bright green). Coordination interactions between backbone carbonyls and potassium are marked with red dotted lines. H-bond interactions between ADPNP and K435 and N48 are marked with gray dotted lines. CryoEM density for ADPNP, metals, and the K435 side chain is shown in semitransparent gray.

as endonucleases in the bacterial MksBEFG/Wadjet plasmid-defense factors, as well as Gabija and other anti-phage systems[19–23].

Biochemical studies have shown that Top6 preferentially interacts with DNA juxtapositions (crossovers) in supercoiled and/or catenated substrates and also actively bends duplex DNA in concert with ATP-coupled B-subunit dimerization[24,25]. Work with Top6 mutants has identified amino acids on two exterior portions of Top6B (the transducer and H2TH domains, Fig. 1a) that are important for ATPase stimulation and recognizing bent DNAs[24], but how these residues support enzyme function has not been defined. How Top6A recognizes and cleaves DNA is similarly unclear. Previous Top6 structures without DNA indicated that the conformational ground state of dimerized Top6A is incapable of cleaving DNA, as each catalytic tyrosine is positioned too far from the $Mg^{2+}$-ion binding site of its partner-subunit's TOPRIM fold and the predicted position of the phosphodiester backbone to promote strand scission[13]. It has been proposed that a conformational change involving the pivoting of the WH domains and tyrosines toward the TOPRIM and DNA is needed for cleavage to take place[13,15]. Recent computationally derived models of mouse SPO11 in conjunction with its Top6B-like partner (Top6BL) have suggested that, like Top6, this system also engages bent DNA substrates and requires an inward pivot of the WH domain to position its catalytic tyrosine close to the phosphodiester backbone[26,27]. Thus far, a fully functional Top6 holoenzyme (or SPO11-TOP6BL heterotetramer) has yet to be imaged with DNA to test these predictions. How the ATPase status of the B subunits might respond to DNA and communicate with the A subunits to promote cleavage also is unresolved.

To advance our understanding of Top6 and its evolutionarily related counterparts, we used single particle electron cryo-microscopy (cryoEM) to determine structures of *Methanosarcina mazei* Top6 with negatively supercoiled DNA minicircles and the non-hydrolyzable ATP analog, ADPNP, in cleaved and uncleaved states. Comparison of five different holoenzyme reconstructions revealed functional elements and coordinated structural rearrangements that mediate DNA recognition, DNA cleavage, and enzyme activity. In binding DNA, Top6 associates with a 74 bp duplex segment bent into a loop akin to that formed at the tips of plectonemes, helping explain why the enzyme prefers binding to supercoiled as compared to linearized substrates. The structures reveal functional elements shown to be important for catalytic activity, including a protein latch that helps anchor Top6A to

DNA, amino acid contacts along Top6A and Top6B subunits that both read out and support DNA bending, an electrostatic clasp that controls WH/TOPRIM interactions to regulate DNA cleavage propensity, and a lever arm on Top6B whose helical stability is necessary for strand passage. Collectively, these findings help explain how type IIB topoisomerases recognize and respond to physiological DNA states while restricting unwarranted cleavage events, insights that advance our understanding of related TOPRIM-dependent systems.

## Results

### Mechanism of DNA binding and bending by Top6

To define the physical basis for DNA recognition by Top6, we took advantage of prior findings showing that the enzyme prefers to bind substrates that are sharply bent or that have crossovers[24,25]. To minimize the technical challenges of using supercoiled plasmids for single-particle cryoEM analyses, we employed, depending on reagent availability, either 306 or 327 bp duplex DNA minicircles (mcDNAs)[28,29]. After generating and purifying the minicircles ("**Methods**"), we then confirmed that they are supercoiled and can be relaxed by *M. mazei* Top6 in an ATP-dependent manner (Supplementary Fig. 1b).

We next prepared and imaged complexes of full-length Top6 and mcDNA either with or without ADPNP using single particle cryoEM. The assembly without nucleotide proved too heterogeneous to obtain useful 2D classes or 3D reconstructions and so was not pursued further. By contrast, data collected for the nucleotide-bound complex were of sufficient quality to obtain a 3D reconstruction at a reasonable resolution (3.7 Å) ("**Methods**", Supplementary Figs. 2, 3). The density maps clearly resolved two Top6A and two Top6B subunits, as well as a highly curved duplex DNA segment (Fig. 1b). The maps allowed for the building of nearly all residues in four protein chains (Top6A$_2$B$_2$), along with 74 bp of DNA and two molecules of ADPNP (Supplementary Table 1). The overall structure is two-fold symmetric, even though symmetry was not imposed during the 3D reconstruction workflow.

In the holoenzyme model, the two Top6B subunits are dimerized (Fig. 1b, c), creating an extensive buried interface between the GHKL, H2TH, and transducer regions (2700 Å$^2$ surface area per protomer) that permits the reciprocal domain swapping of a small portion of the extreme N-terminus (the 'strap') of each subunit with its partner ATPase fold (Supplementary Fig. 4a). This state is akin to prior structures of the isolated Top6B subunit with nucleotide[14,30] but is unlike

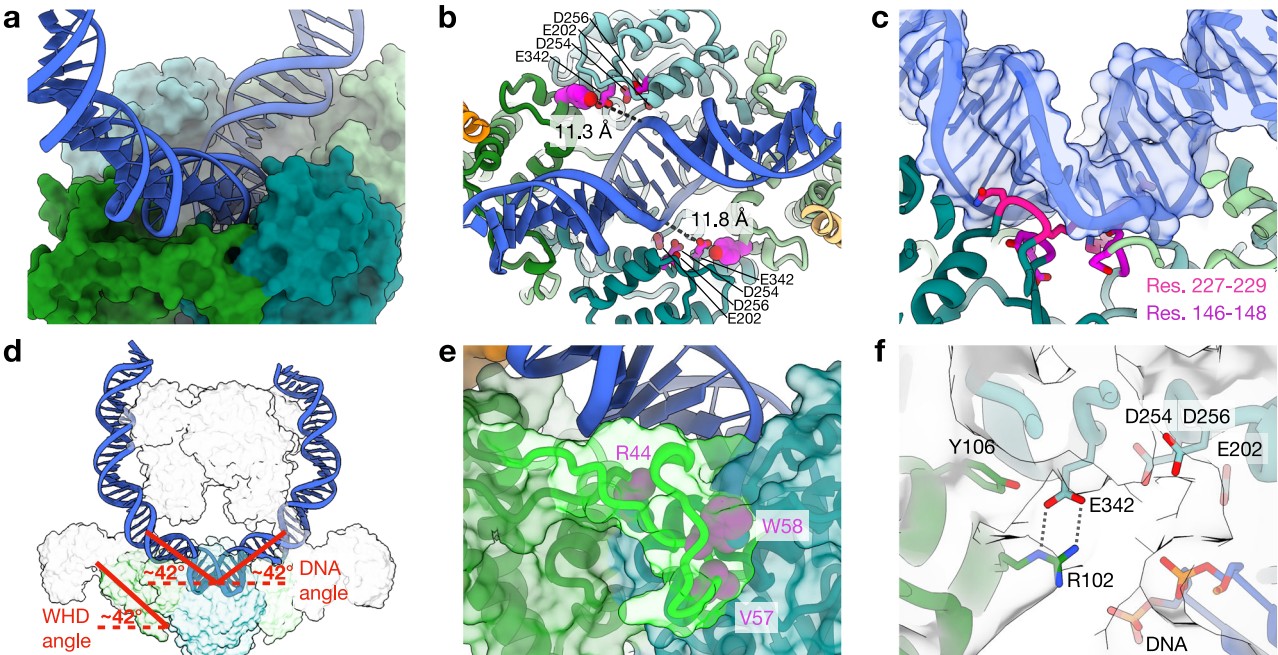

**Fig. 2 | DNA-binding and active site features of Top6A. a** Oblique surface view of the DNA-binding channel formed by the A-subunit dimer with bound mcDNA rendered as a cartoon. **b** Top-down view of the DNA-binding channel with the following active-site constituents shown in atomic representations (magenta): Y106 (spheres, depicted along with the distances between the OH group and the DNA backbone phosphate) and the acidic tetrad of E202, D254, D256, and E342 (sticks). **c** Two loops from the TOPRIM domain (magenta and purple) bind to and deform the DNA minor groove. Residues that contact the duplex are shown as sticks. **d** View depicting the DNA bend and angle, measured as degrees deviated from a straight duplex ("**Methods**"). The WH domain position is measured as the angle of helix 1 from the horizontal. **e** Backbone and surface representations of the WH domain latch, colored bright green, in its surrounding context, with three key packing residues shown as labeled magenta spheres. **f** Close up of active site-related residues (sticks), including the catalytic tyrosine (Y106), an electrostatic clasp (R102/E342), and TOPRIM metal-binding residues (E202/D254/D256) in an uncleaved (metal free) DNA-bound state. The cryoEM density map is shown in semi-transparent gray. A salt bridge between R102 and E342 is shown with gray dotted lines.

those in nucleotide- and DNA-free holoenzyme configurations, in which the ATPase regions were either fully separated from each other or not as closely packed together[15,31]. Each GHKL ATPase fold associates with an ADPNP molecule and a $Mg^{2+}$ ion that is coordinated by the nucleotide triphosphate moiety and the N48 side chain of Top6B; the γ-$PO_4$ group of ADPNP is further contacted by K435, which sits on a loop provided by the transducer domain (Fig. 1d). A second metal ion that had not been noted in prior structures (likely $K^+$ based on coordination geometry and distances[32]) is also seen in each Top6B subunit, interacting with the backbone carbonyl oxygens of R102, A105, K107, and I426 (Fig. 1d). Although this ion interacts with a triphosphate-binding loop known as the 'ATP lid' in GHKL ATPases, the liganding differs from a known $K^+$-ion binding locus in the ATP lid of other GHKL relatives[33]; rather it serves to anchor the lid to the transducer domain (Supplementary Fig. 4b). The remainder of the Top6B sequence, an immunoglobulin β-sandwich domain that is found at the far C-terminus of certain Top6 orthologs (including *M. mazei*), and which may help localize Top6 to a partner protein or subcellular region[15], is evident only at low threshold and is poorly ordered, indicating that its position is dynamic.

As with Top6B, both A subunits of Top6 also form a dimer (Fig. 1b, c), whose conformation is reminiscent of that seen for the isolated Top6A subunit and for nucleotide- and DNA-free Top6 holoenzyme states[13,15,31]. The C-terminal TOPRIM folds abut and are flanked by the N-terminal WH domains, each of which contacts both their own and their partner's TOPRIM element (Supplementary Fig. 5a). The Top6A WH domains also bind to a small C-terminal helical hairpin in Top6B, which follows a long α-helical stalk that extends from the transducer domain (Supplementary Fig. 5b). Collectively, the WH and TOPRIM domains form a deep channel in the Top6A dimer that is occupied by a fully intact DNA duplex (Figs. 1c, 2a). The catalytic tyrosine (Y106) of

each WH domain sits far (~10 Å) from the acidic tetrad of its dimer-associated TOPRIM fold (E202/E254/D256/E342), and no metal ion is evident in this region, indicating that the observed state corresponds to a nucleotide-associated, pre-cleavage or post-religation intermediate (Fig. 2b).

The DNA in the holoenzyme structure is strongly bent (to >180°) as it passes through the A-subunit channel, a deformation that is accentuated by contacts with both Top6Bs to arch upward and partially over the holoenzyme (Figs. 1c, 2a). DNA curvature is enforced by two loops (residues 146–148 and 227–229) from each Top6A subunit that insert into the DNA minor groove adjacent to the point where the duplex passes over the dimer axis, creating an 84° local bend (Fig. 2c, d). Several positively charged side chain clusters from the H2TH domain and helical stalk of Top6B support these interactions, as do a mix of charged and van der Waals contacts from the Top6A channel (Supplementary Fig. 5c). Overall, the degree of DNA bending resembles that expected at the apical tip of a plectonemic supercoil[34] or the sharp kink of a non-plectonemic supercoiled minicircle[35], helping to explain why Top6 shows a preference for binding supercoiled substrates. Top6 was not seen bound to a supercoiled DNA crossover in the EM data, possibly because the addition of ADPNP promoted a single-turnover strand passage event that removes such structures.

One unforeseen consequence of binding to supercoiled DNA is that clear density is now evident for residues 44–67 of the WH domains, which have been disordered in prior Top6A (and Spo11) structures. This region now forms an ordered, long loop that extends from the WH domain in each monomer to its associated TOPRIM fold (Fig. 2e). Several interactions are established in this contact site, including salt bridges and hydrophobic side chain packing associations. In addition, a portion of this loop (along with a small segment of the TOPRIM fold, residues 270–273) also threads through the major

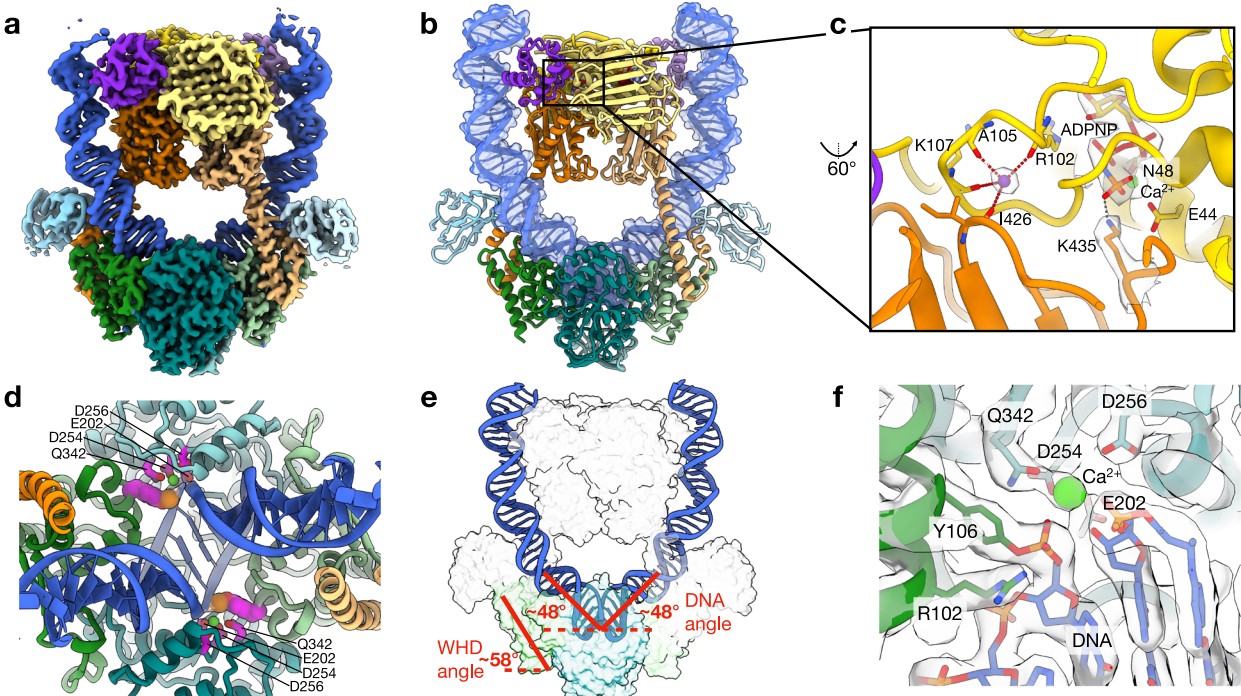

**Fig. 3 | Overview and active site features of the Top6$^{(A:E342Q)}$•mcDNA•ADPNP cleavage-complex structure. a** CryoEM density map of the complex (EMD-49972). **b** Cartoon representation of the atomic model derived from the map (PDB ID: 9OOG). The DNA is shown with a semitransparent atomic surface. **c** Detail of the ADPNP- and metal-binding region of GHKL and transducer domains in a B-subunit monomer. CryoEM density for ADPNP, metals, and the K435 side chain is shown in semitransparent gray. **d** Top-down view of the cleaved DNA with the following active-site constituents shown in atomic representations (magenta): phosphotyrosine (residue 106 and the DNA backbone phosphate) (spheres), the metal-binding tetrad (E202, D254, D256, and Q342, sticks); and calcium ion (bright green sphere).

**e** View of the DNA bend and angle, measured as degrees deviated from a straight duplex ("**Methods**"). The WH domain position is measured as the angle of helix 1 from the horizontal. **f** Close up of active site constituents in the cleavage state, including the catalytic tyrosine (Y106), the electrostatic clasp (R102 and Q342, mutated from glutamate), the TOPRIM metal-binding site (E202/D254/D256), and the associated calcium ion. The cryoEM density map is shown in semitransparent gray. Distances from the carboxylic oxygens of E202 and D256 to the associated metal are too great to represent inner shell contacts (~4 Å) and may be mediated by water molecules that are not evident at the overall resolution of the Coulomb potential maps.

groove of the bound DNA, stabilizing the bending of the DNA toward the Top6B subunits (Fig. 2e). The disorder/order transition of the loop along with its placement—appearing to latch the WH domain to both the DNA and its downstream TOPRIM region—suggest that the element is functionally important. Moreover, inspection of a phylogenetic analysis revealed that this loop is conserved in length and at key amino acid positions in archaeal Top6A and vertebrate Spo11 proteins, while being more variable in other eukaryotic Spo11 homologs (Supplementary Fig. 6a)[36]. To test the relevance of the loop, we mutated three of the conserved amino acids that make substantive inter-molecular contacts with the TOPRIM fold (R44, V57, and W58, all to alanine) and compared the supercoil relaxation activity of the mutant Top6 holoenzyme to wild-type Top6. The triple mutant showed no activity in this assay (Supplementary Fig. 6b), indicating that the integrity of the latch is critical for function.

## Top6 cleavage-complex formation

The cleavage-religation equilibrium for topoisomerases is typically shifted toward religation[37–39]. In type IIA topoisomerases, mutations have been found that shift the equilibrium toward the cleaved state[40–42]. These alterations frequently involve salt bridges that map to inter-domain or inter-subunit contact points[43]. To understand how Top6 transitions from an uncleaved to cleaved conformation, we inspected the interface between the WH and TOPRIM domains for highly conserved ion pairs across Top6A and Spo11 homologs and noted one, formed between the side chains of residues E342 and R102, that appeared to restrain the WH domain catalytic tyrosine from

reaching the TOPRIM acidic triad (Fig. 2f and Supplementary Fig. 7a). To discern the role of this prospective electrostatic clasp, we mutated E342 to Q in Top6A, expressed and copurified the mutant subunit with Top6B, and performed a minicircle cleavage assay. Top6$^{(A:E342Q)}$ proved much more prone to generating DNA breaks than the wildtype enzyme (Supplementary Fig. 7b), establishing that the clasp regulates cleavage.

To exploit and optimize the DNA strand breakage properties of Top6$^{(A:E342Q)}$, we also tested for cleavage activity with different divalent cations. $Ca^{2+}$ was found to yield higher dsDNA cleavage activity than $Mg^{2+}$, which proved more prone to supporting DNA nicking (Supplementary Fig. 7b). Using this information, we next prepared cryoEM grids with Top6$^{(A:E342Q)}$, a 306 bp minicircle ("**Methods**"), ADPNP, and $CaCl_2$. CryoEM data collection and 3D reconstruction efforts produced maps that extended to ~2.5 Å resolution overall (Supplementary Figs. 8, 9), allowing for the building and refinement of a two-fold symmetric model (without imposed symmetry) that includes all residues except for a few N- and C-terminal amino acids in each subunit, two molecules of ADPNP, and 74 bp of mcDNA (Supplementary Table 1 and Fig. 3a, b). As with the uncleaved model, $Ca^{2+}$ and $K^+$ were seen associated with each ADPNP and secondary divalent ion-binding site in Top6B, respectively; in this state, the divalent cation is coordinated by the side chains of both E44 and N48 (Fig. 3c).

Unlike the uncleaved state, the reconstructed map for the Top6$^{(A:E342Q)}$ construct shows that the substrate DNA that passes through the A-subunit channel is clearly broken. The overall path of the DNA is essentially identical to that in the intact, uncleaved structure but is now more sharply kinked (~96° overall) at the center of the

A-subunit dimer (Fig. 3d–f and Supplementary Fig. 10a). In both Top6A active sites, the density is consistent with the formation of a single phosphotyrosine linkage between the 5′ end of each DNA strand and Y106 (Fig. 3f), in agreement with prior predictions that this residue serves as the catalytic side chain that attacks DNA[3,18]. The cuts are staggered, resulting in 2-base overhangs, confirming the findings of an earlier biochemical analysis using *S. shibatae* Top6[44]. Each active site is bipartite, formed by an ~16° inward pivot (measured as the pivot of helix 1 from an origin at its N terminus) of the WH domain of one subunit from its resting position in the uncleaved state to engage the TOPRIM fold of its partner (Fig. 3d, e and Supplementary Movie 1). Within the TOPRIM fold, density is evident for a single $Ca^{2+}$ ion coordinated by side chains D254, D256, E202, and Q342, as well as the phosphate of the phosphotyrosine, and the adjacent ribose 3′-OH of the cleaved DNA strand (Fig. 3f). In this state, the 3′-OH of the broken DNA is close to (3.3 Å) but misaligned (~30–40° offset) with respect to the central phosphorous atom of the phosphotyrosyl linkage. This configuration may be stabilized by the different liganding capacity of the $Ca^{2+}$ ion (as opposed to $Mg^{2+}$) that associates with the 3′-OH and/or the contact formed between the ion and the amide oxygen of Q342, an interaction that would be replaced with a carboxylate in WT Top6. Overall, the formation of the broken DNA ends alongside a metal-coordinated phosphotyrosine intermediate establishes the structure as a post-cleavage or pre-religation state of the enzyme.

## Top6 cleaves DNA at a sequence-encoded deformability motif

We had expected the observed DNA density in both Top6 states to reflect an averaged sequence because the enzyme could, in principle, engage many different sites in the supercoiled minicircle substrate. Surprisingly, the resolution in the central ~28 base pair region of the cleaved state was sufficiently high (~2.5 Å resolution) to permit an assessment of sequence specificity using a new, ad hoc base identification method termed Base Hunting[45] (Baker et al., 2025, is currently a preprint and has not been peer reviewed by a journal). This method uses shape-based descriptors to classify individual nucleotide density as a purine (R), a pyrimidine (Y), or as undetermined and creates an R/Y profile based on observed structural features without prior knowledge of the DNA sequence. The structure-based profile was aligned with the known R/Y sequence from the minicircle to scan for a prospective binding register. R/Y assignments were clear for 26 of 28 base pairs in that region; of these, 24 aligned perfectly with a single site match on the negatively supercoiled minicircle (Supplementary Table 2). This finding establishes that Top6 associates with a specific region of the mcDNA in the cleavage state (Supplementary Fig. 10b).

While it was known that Top6 recognizes bent DNA[24] and that minicircles can have sharp bends[35], it was initially unclear why Top6 would preferentially bind and cleave a particular DNA sequence in the supercoiled minicircle, particularly one that does not bear the hallmarks of a DNA breakage hotspot (AT-rich overhangs) noted previously[44]. DNA sequence is known to significantly impact the ability of DNA to be bent and/or compressed[46,47]. Using DNA-protein crystal structures, the inherent deformability of specific DNA sequences has been described numerically by the average volume taken up for each unique base pair step[47]. This approach is based on the idea that base pair steps that are easier to deform into a wider range of shapes take up more volume on average. Sequence-dependent deformability values have been described for individual base pair steps[47]; however, it has been less evident how such values change as a function of DNA segment length. Reasoning that the propensity of a given DNA sequence to bend and deform more readily than others might contribute to the observed binding site preference of Top6, we applied a newly developed computational tool[45] to scan the sequence of the negatively supercoiled minicircle substrate for patterns of DNA sequence-dependent deformability in the Top6 binding site. This analysis

showed that while the ~30 base pairs centered around the cleavage site only fall within the 58th percentile of the most deformable 30-mer sequences in the 306 bp mcDNA, the sequences immediately adjacent to the cleavage site partition into two halves with distinctly opposing deformabilities: the 30 bp immediately upstream from the cleavage site are highly non-deformable (deformability score falling into the lowest 15th percentile) whereas the 30 base pairs immediately downstream from the cleavage site are highly deformable (95th percentile) (Supplementary Fig. 10c, d). In addition, the two base pair steps that were cleaved by Top6, the CG in the tetramer ACGA (forward strands, chains H/I) and the CT in the tetramer GCTC (reverse strand, chains F/G), both share an above-average deformability value of 4.6 degrees$^3$•Å$^3$. There is only one other region of the mcDNA that has a similar pattern, which is located on the opposite side of the mcDNA from the cleavage site (Supplementary Fig. 10c), but the central tetramer in this segment lacks the symmetric above-average deformability exhibited by the cleavage site. Thus, while our current model of deformability is unlikely to capture all of the biophysical properties of the mcDNA, it nonetheless indicates that Top6 appears to preferentially bind and cleave at the intersection between highly deformable and relatively rigid DNA segments, particularly when this intersection is itself deformable. This bias may arise because several of the base pairs around the cleavage site, particularly on the more deformable arm, are warped from standard B-form DNA into a more A-like configuration (Supplementary Fig. 10e). The apparent reliance on DNA flexibility for cleavage site selection by Top6 is notable, as it parallels a similar dependency that has been recently found for *E. coli* gyrase, a type IIA topoisomerase[45].

## Protein flexing accompanies dynamic contacts with DNA arms that regulate Top6 function

Outside of the central channel of the Top6A dimer, the remainder of the DNA contacts are with the outer edges of Top6B. Some of the observed contact points—both a region of the C-terminal stalk bearing a conserved WKxY motif and the H2TH domain (residues R399/K401, and R263/K268/K308, respectively)—confirm prior predictions derived from biochemical and mutagenesis analyses[24]. However, the specific molecular interactions between these two regions of Top6B and supercoiled mcDNA are now apparent, as are three additional binding loci: one near the N-terminal end of the transducer stalk (residues R457/K458/K460/H461), one in the middle (K465), and one near the C-terminal end (K471/K475) (Fig. 4a). Intriguingly, the Top6B-DNA arm contacts of the symmetric cleaved and uncleaved states are similar but not identical. Moreover, upon further classifying and refining our cryoEM data, we identified two additional states for the WT protein and one for the E342Q mutant (Supplementary Table 1 and Supplementary Fig. 11a). These states proved highly asymmetric, a physical distinction that led several of the DNA contacts to also differ from one state to another, even between dimer-related protomers from the same reconstruction (Table 1, Fig. 4b and Supplementary Fig. 11b, c).

While the resolution of the asymmetric states is generally lower than the symmetric forms, the cryoEM maps nonetheless clearly show that the Top6B transducer domains and stalks undergo considerable conformational flexing (Supplementary Movie 2). For the asymmetric state of Top6$^{(A:E342Q)}$, both transducers undergo a coordinated tilting movement to give rise to a lopsided particle with a squat, rounded curvature for the bent DNA (Supplementary Fig. 11b). Despite this motion, the WH domains remain rotated inward toward each other, the central DNA segment is still kinked, and the transducer stalks are relatively straight (the resolution for this state was insufficient to resolve whether Y106 has cleaved the phosphodiester backbone, but the refined distance is more consistent with a hydrogen bond than a covalent linkage). By comparison, two different asymmetric states found for WT Top6 correspond to a lopsided form with a wide gap

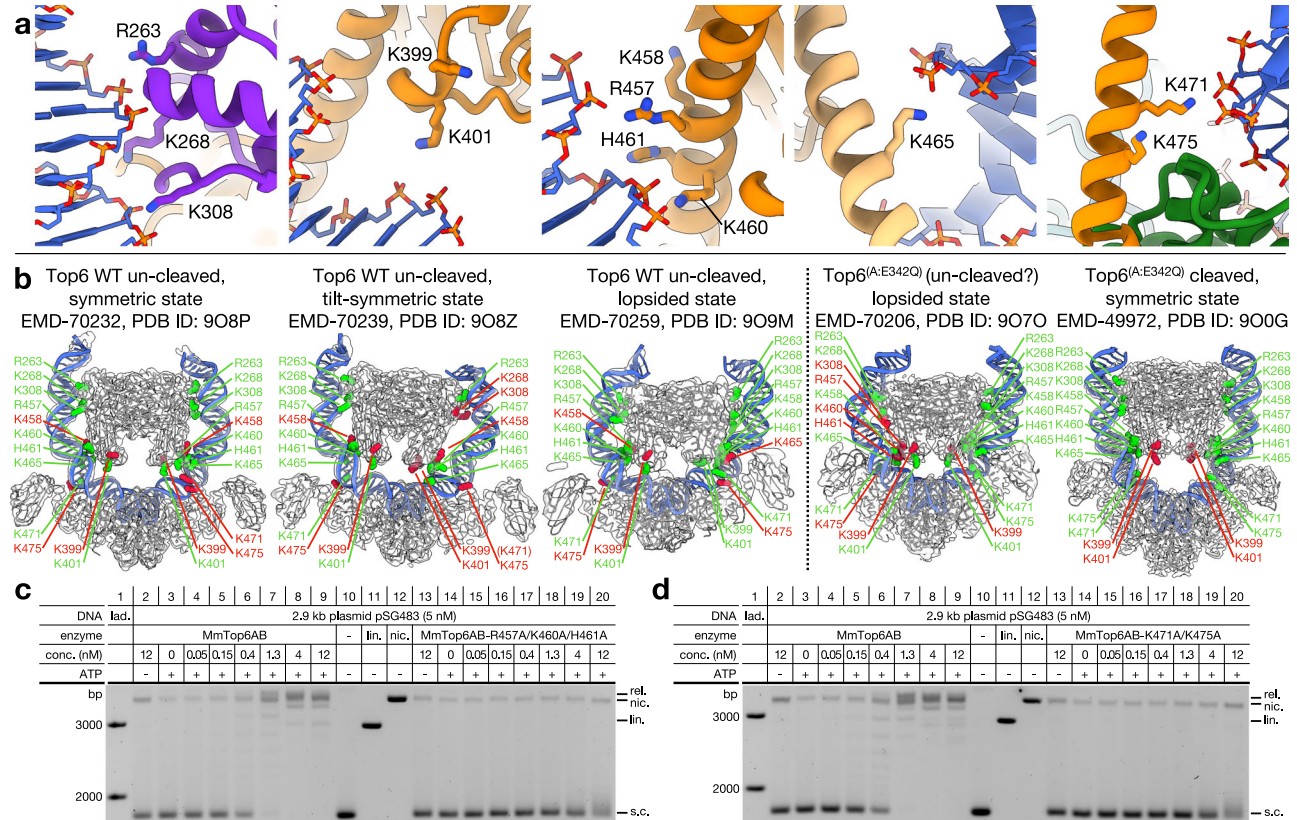

**Fig. 4 | B-subunit/DNA contacts and the conformational plasticity of Top6.**
**a** Examples of H2TH (purple cartoon) and transducer (orange cartoon) domain interactions with DNA (blue cartoon with orange/red stick phosphodiester backbone) in the symmetric, uncleaved state. **b** Comparison of the five different Top6 holoenzyme reconstructions determined in this paper (WT symmetric: 9O8P/EMD-70232, WT tilt-symmetric: 9O8Z/EMD-70239, WT lopsided: 9O9M/EMD-70259, Top6(A:E342Q) lopsided: 9O7O/EMD-70206, Top6(A:E342Q) cleavage: 9O0G/EMD-49972), shown as thin cartoon models in semitransparent gray cryoEM density maps. The

DNA-binding residues of the B subunits are shown as spherical atomic representations and colored and labeled in green or red depending on whether they are close enough to engage DNA in that state. **c**, **d** Native agarose gels of negative DNA supercoil relaxation activity comparing the Top6(B:R457A/K460A/H461A) and Top6(B:K471A/K475A) transducer stalk variants, respectively, with the wildtype enzyme. Results are shown from 1 of 2 virtually identical experimental replicates for each panel. Uncropped gel images are provided as a Source Data file.

## Table 1 | DNA-binding residues engaged in different states

| | WT Top6 symmetric | WT Top6 tilt-symmetric | WT Top6 lopsided | Top6(A:E342Q) lopsided | Top6(A:E342Q) symmetric (cleavage) |
|---|---|---|---|---|---|
| **H2TH** | R263 | R263 | R263 | R263 | R263 |
| | K268 | K268 B side only | K268 | K268 | K268 |
| | K308 | K308 B side only | K308 | K308 D side only | K308 |
| **WKxY** | | | K399 D side only | | |
| | K401 | K401 B side only | K401 | K401 | R457 |
| | R457 | R457 | R457 | R457 D side only | |
| **Additional transducer residues** | | | K458 D side only | K458 | K458 |
| | K460 | K460 | K460 | K460 D side only | K460 |
| | H461 | H461 | H461 | H461 D side only | H461 |
| | K465 | K465 | K465 B side only | K465 | K465 |
| | K471 B side only | K471 B side only | K471 | K471 | K471 |
| | | | | K475 D side only | K475 |

between transducer domains and a near-symmetric form that exhibits a highly tilted ATPase dimer, a narrow transducer domain gap, and a single unfolded transducer stalk (Supplementary Fig. 11c). In both asymmetric wildtype states, the WH domains are rotated outward from the TOPRIM regions and the mcDNA in the Top6A channel is smoothly bent (Supplementary Fig. 11d), consistent with the presence of an uncleaved substrate in these conformations.

The complex repositioning of the transducer and WH domains between Top6 states and the shifting landscape of Top6B-DNA arm interactions prompted us to quantify the movement of these two domains in comparison with four other defined elements of the holoenzyme (GHKL ATPase, H2TH, Top6B CTD, and TOPRIM) (Supplementary Table 3). The GHKL domain exhibited the greatest overall conformational variability, followed by the H2TH and transducer

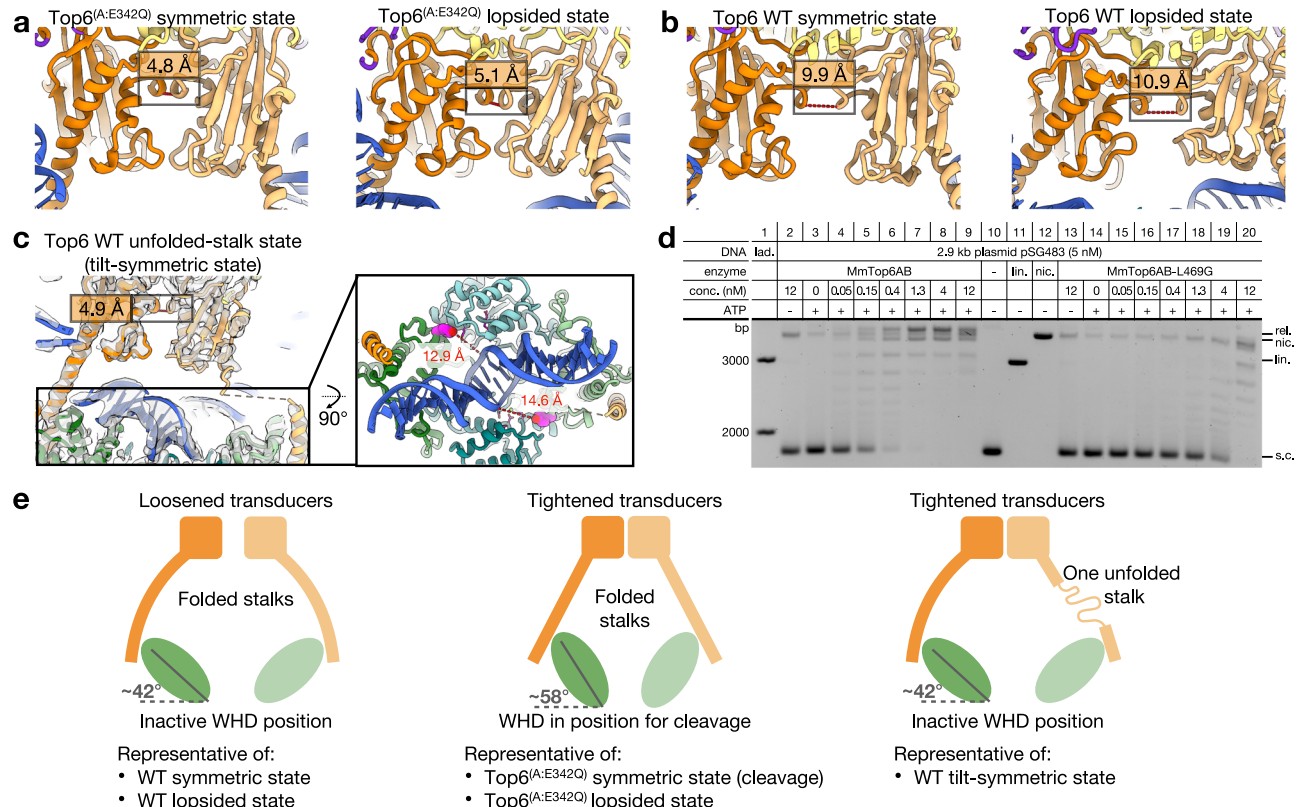

**Fig. 5 | Transducer domain status across the five states of Top6 and the role of stalk stability on enzyme activity. a** Relative positions of the transducer domains (dark and light orange cartoon) in the Top6$^{(A:E342Q)}$ states, in which both transducer stalks are folded (cleavage state, 9O0G; lopsided state, 9O7O). The distance between the G387 Cα atoms, which are directly juxtaposed across the dyad between B subunits, is used as a proxy for transducer tightness. **b** Relative positions of the transducer domains (dark and light orange cartoon) in the two WT states in which both transducer stalks are folded (symmetric state, 9O8P; lopsided state, 9O9M). **c** Positions of the transducer domains in the tilt-symmetric state, in which

one stalk is unfolded (9O8Z). CryoEM density (EMD-70239) is shown in semi-transparent gray. Inset (at right) shows a top-down view of the DNA binding channel in this state, with the distances between Y106 (magenta spheres) and DNA backbone annotated. **d** Native agarose gel showing DNA supercoil relaxation by the Top6$^{(B:L469G)}$ transducer stalk variant as compared to wildtype Top6. Result shown from 1 of 3 distinct experiments. An uncropped gel image is provided as a Source Data file. **e** Simplified illustration of the relationship between transducer status and WHD position.

regions, while movements of the WH and TOPRIM domains were relatively restricted, consistent with their role anchoring DNA and bridging the DNA-gate dimer. Correlation analysis revealed that local DNA bending is tightly linked to conformational changes across specific Top6B domains (Supplementary Tables 3, 4 and Supplementary Fig. 12). The GHKL and H2TH domains undergo larger positional shifts in states where the DNA is sharply kinked, whereas the TOPRIM and CTD domains become more flexible as DNA deformation eases. H2TH-DNA interactions persist throughout these movements and are like those seen in other proteins that contain this element, such as MutM/Fpg glycosylases, EndoVIII, and human EndoVIII-like (NEIL1), where they are used for DNA backbone recognition and sometimes DNA deformation (reviewed in Hitomi et al., 2007[48]). This consistency suggests that the H2TH domain serves not merely as a passive anchor but as a responsive interface that tracks and potentially transduces DNA deformation into conformational change within Top6B.

We next sought to define the importance of the observed Top6B-DNA arm contacts to overall Top6 function. The H2TH and WKxY interactions have been tested previously and are required for supercoil relaxation activity[24]. To assess the impact of the dynamic interaction points, we created alanine substitutions in two clusters (R457A/K460A/H461A and K471A/K475A) and tested the supercoil relaxation activity of the mutant Top6 proteins. Each set of substitutions substantially reduced supercoil relaxation activity (at least 25-fold) compared to the WT enzyme (Fig. 4c, d). Thus, the full array of contact points that Top6B uses to associate with negatively supercoiled DNA

are critical for overall function, even though they do not always maintain these interactions as the enzyme conformationally flexes.

### Top6B transducer stalk stability appears to coordinate ATPase motions with DNA cleavage by Top6A

Even though it is dimerized in all observed states, the flexion seen for the Top6 ATPase was unexpected. Conformational shifts between the GHKL and transducer domains (which control the interactions between a conserved lysine on the transducer and the γ-phosphate of ATP) have been seen for the isolated Top6B subunit and in DNA-free Top6 holoenzyme structures[15,30,31,49], but not in a defined ensemble such as that imaged here for the DNA-bound states. For the two states formed by the Top6$^{(A:E342Q)}$ cleavage-prone mutant, the transducer regions are pinched together (toward the GHKL dimer axis) to form a tightened disposition (Fig. 5a). This configuration is accompanied by well-folded, straight transducer stalks and an inward pivot of the Top6A WH domains that brings the catalytic tyrosines into proximity with DNA. By comparison, in two of the uncleaved WT Top6 states—symmetric and lopsided—the transducer domains are splayed away from each other in a loosened conformation and the WH domains and catalytic tyrosines are pivoted outward, away from DNA (Fig. 5b). Collectively, these correlations suggest that cleavage competency by Top6A is directly coupled, at least in part, to inter-transducer status in Top6B.

An inspection of the tilt-symmetric state of WT Top6A unexpectedly indicated that the control of DNA cleavage might also be

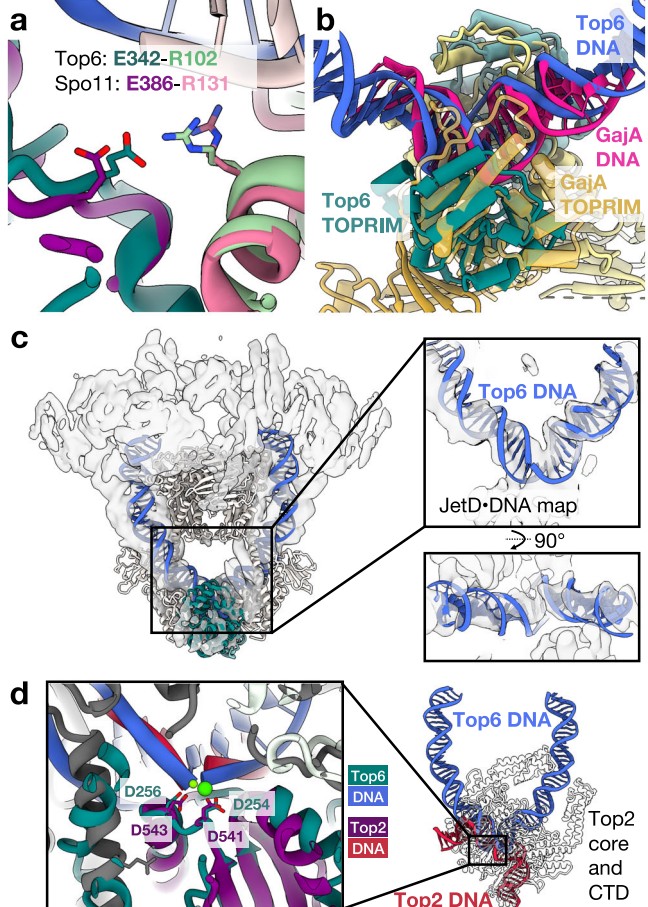

**Fig. 6 | Comparisons of Top6 with other TOPRIM-based DNA cleaving enzymes.**
**a** The electrostatic clasp residues seen in Top6 are present in Spo11. A hypothetical model by Yu et al., 2025, of an *S. cerevisiae* Spo11 dimer bound to DNA[52] is superposed with the uncleaved Top6-mcDNA model determined here. Spo11 is colored dark magenta (TORPIM), pink (WHD), and light pink (DNA), while the equivalent regions of Top6 are dark teal, sea green, and blue. **b** TOPRIM fold-based superposition of GajA bound to uncleaved DNA structure[53] (PDB ID: 8X51) on the symmetric, uncleaved state of Top6 bound to mcDNA. GajA is colored yellow and its DNA bright pink. **c** Top6[(A:E342Q)] cleavage state model (colored light gray overall, with teal TOPRIM domains and royal blue DNA) docked into JetD-focused refined map of the JetABCD complex[54], colored semitransparent gray. The conformation of the cleaved, Top6-bound DNA closely matches that bound to JetD. **d** Superposition of the human Top2α cleavage core[55] (PDB ID: 4FM9) with the cleaved state of Top6, aligned using the DxD motifs of their respective TOPRIM domains. In the detail panel (left inset), the TOPRIM of the Top2 structure is colored purple, and the DNA is colored red. In the overview (right), the orientation of the DNA in the Top2 complex is shown relative to that of Top6. The Top6 protein backbone is not shown for visual clarity, while the Top2 catalytic core is shown as a semitransparent pale gray cartoon.

linked to transducer stalk stability. Here, the transducer dimer configuration is tight, as with the cleavage mutant, but the WHDs are pulled back into an uncleaved position seen in the other WT Top6 forms. Moreover, the density for the middle region of one of the Top6B transducer stalks has disappeared, and the distance between the ordered helical ends is too great to accommodate an α-helix, indicating that this segment has become unfolded (Fig. 5c). This disruption suggested that the folding state of the stalk might help couple transducer status to the ability of Top6A to attain a cleavage competent conformation. To test this idea, we decreased the stability of the transducer stalk by substituting a single helix-favoring residue (L469) in the middle of the region with glycine[50]. We then tested supercoil

relaxation by this mutant compared to WT Top6 and found that it was impaired by at least 10-fold (Fig. 5d). Thus, the stalk appears to serve as a tension sensor in which helical integrity helps the transducer domain position the associated WH domain to promote DNA cleavage (Fig. 5e).

## Insights into related systems possessing TOPRIM domains

The TOPRIM domain is ancient and widespread. TOPRIM folds support RNA synthesis by DnaG-family primases, DNA cleavage by type IA, IIA, and IIB topoisomerases, DNA repair and recombination proteins (*e.g.*, RecR), meiotic recombination (Spo11), and phage and plasmid defense factors (MksBEFG/Wadjet, Gabija, and OLD systems)[17,51]. Comparisons of supercoiled DNA-bound states of Top6 obtained here with other proteins that utilize the TOPRIM domain highlight congruent architectural features and functional properties, particularly for those factors that cleave DNA. A recent cryoEM structure of undimerized budding yeast Spo11 with its B-subunit equivalent (Rec102/104) shows a path for DNA along the TOPRIM fold that closely matches that seen here for Top6[52] (Supplementary Fig. 13a). Although the relative orientation of the WH domain is fairly different between the two structures, the configuration seen for this domain with respect to its adjoining TOPRIM fold in Top6 is closely predicted by a model of the Spo11 dimer that was adapted from the monomer cryoEM structure[52] (Supplementary Fig. 13b). Moreover, AlphaFold-based modeling studies with mouse SPO11 have suggested that the dimerized form of the complex bends DNA toward its B-subunit counterpart (TOP6BL) in a manner that closely matches what is seen for Top6[27], predicting several protein-DNA contacts that are confirmed here (Supplementary Fig. 13c, d). Finally, sequence and structural alignments have shown that the electrostatic clasp is highly conserved between Top6A and Spo11, indicating that both systems use this linkage to restrict inappropriate DNA cleavage (Supplementary Fig. 7a and Fig. 6a). Overall, these congruencies reinforce prior suggestions that the DNA recognition and strand breakage mechanisms of Top6 and Spo11 are closely aligned with one another[3,15,18,26,27,52].

The DNA deformations observed for Top6 are seen in other TOPRIM domain-containing proteins as well. For example, an overlay of uncleaved DNA complexes between Top6 and GajA (a Gabija-family nuclease[53]) shows the two proteins bend their target DNA duplexes to a similar degree (Fig. 6b). Comparable DNA bends are also seen between the cleaved-DNA state of Top6 and a catalytically inactive mutant of JetD (an MksBEFG/Wadjet-family nuclease) bound to plasmid DNA and the JetABC loop-extrusion motor (Fig. 6c); the sharper degree of bending seen in these two instances as compared to the uncleaved DNA states (~ 95–100° vs ~ 84°) is consistent with the proposal that loop extrusion by MksBEFG/Wadjet factors promotes plasmid cleavage when the target DNA attains a sufficiently bent state[54]. Overall, these structural congruencies suggest that TOPRIM-based DNA cleavage enzymes, particularly those in which TOPRIM folds directly about through dimer contacts, may generally use DNA bending to discern appropriate target substrates and regulate catalytic function. Interestingly, in type IIA topoisomerases, the TOPRIM domains do not directly touch each other, and dimerization is instead mediated by WH domain contacts[55] (Supplementary Fig. 13e). Consequently, the DNA bound by these enzymes bends in the opposite direction as seen in Top6 (Fig. 6d), a marked physical divergence that reflects the reshuffled domain architecture of the two protein families[13,56].

## Discussion

Top6 preferentially binds to and is stimulated by supercoiled DNA, and its reaction cycle depends on ATPase dimerization for DNA cleavage and strand passage[14,24]. Early studies hypothesized that Top6A would need to sample multiple conformations to control DNA cleavage[13–15,24,31], and it has been proposed that ATP-dependent subunit dimerization could constitute one mechanism that restrains inappropriate cleavage events not just by Top6, but by type II

topoisomerases in general[39]. However, the conformational changes and physical linkages that allow Top6 to couple supercoil recognition and DNA cleavage with the ATPase cycle have until now remained poorly defined at a molecular level.

To better understand the nucleotide-dependent control mechanisms that regulate Top6 function, we determined five different holoenzyme conformations of *M. mazei* Top6 with supercoiled DNA minicircles and ADPNP. We compared these states both to each other and to holoenzyme structures without DNA (Figs. 1b, 3a, 4b)[15,31]. The ensemble of states imaged here confirms prior predictions that ATP binding promotes cleavage through GHKL domain dimerization[14,15,24,31], bringing the Top6B transducer elements close together and pivoting the associated Top6A WH domains in toward DNA where their catalytic tyrosines can approach the phosphodiester backbone. However, we discovered that the control of the WH domain position and Top6 activity depends on additional physical elements as well. One is an electrostatic clasp (formed between E342 and R102) that sterically restrains the position of each Top6A WH domain to block its catalytic tyrosine (Y106) from reaching the DNA (Fig. 2f). A second element is the helical stalk that connects the Top6B transducer and Top6A WH domains and which was observed to undergo an order/disorder transition in response to ATPase/cleavage status (Fig. 5a–c). A third is a previously uncharacterized segment in Top6A (residues 44–67) that became ordered upon binding supercoiled minicircle DNA to stretch across the substrate duplex and latch each WH domain to its adjoining TOPRIM fold (Fig. 2e and Supplementary Fig. 6a). Mutation of any of these three elements disrupted normal Top6 function, either making the enzyme more cleavage prone (e.g., the E342Q substitution in the clasp) (Supplementary Fig. 7b) or abrogating supercoil relaxation (latch/stalk alterations) (Supplementary Fig. 6b and Fig. 4c, d). Although the clasp appears to play a key role in allowing wildtype Top6 to maintain an intrinsically low level of cleavage (as artificially releasing it by mutation allowed the cleaved state to be populated and imaged by cryoEM), previous work has shown that T-segment DNA binding elements also play a role in regulating cleavage[24]. DNA dynamics outside of the enzyme may also impose tension on the G-segment that is transduced into the enzyme to alter the stability of the clasp, latch, and/or stalk helix. Even though the GHKL, TOPRIM, and WH domains are conserved, the specific molecular safeguards that Top6 uses to couple the appropriate recognition and cleavage of supercoiled/bent DNA segments with the ATPase cycle are distinct from those used by type IIA topoisomerases.

Our structural studies also reveal important insights into how DNA topology influences Top6 function. The preference of Top6 for decatenation has been reported to involve recognition of crossovers of a particular angle[25], while its preference for supercoiled DNA had been found to involve crossover recognition by T-segment binding residues in the central cavity and G-segment-binding sites on the outer surface of the B-subunit[24]. Our structures establish that Top6 can also recognize sharply bent segments naturally present in supercoiled DNA without binding a crossover. In addition, the molecular contacts formed by these bend-sensing regions include two positively charged clusters in the B-subunit stalks (R457/K458/K460/H461 and K471/K475), in addition to previously described interactions, and two loops in Top6A (residues 146–148 and 227–229) (Figs. 4a, b, 2c). Comparing different models reconstructed from our cryoEM data allowed us to observe that interactions between the bent DNA arms and the B subunits are dynamic and can flip between engaged or disengaged states that correlate with local conformational changes in both the enzyme and DNA (Fig. 4b). Mutagenesis shows that each cluster in this extensive, yet adaptable engagement of bent DNA is necessary for productive DNA strand passage activity, indicating that Top6 possesses a functional plasticity that allows it to respond to the inherent flexible nature of DNA (Fig. 4c, d).

The development of a new tool to discern DNA sequence from EM maps[45] enabled the discovery that DNA sequence-based deformability −in this case, consisting of a moderately deformable central cleavage site sandwiched between highly deformable and rigid segments of DNA−likely controls where Top6 preferentially binds and cleaves DNA while also enabling the formation of lopsided states in which the DNA is asymmetrically bent (Supplementary Figs. 10b–d, 11b, c). Analyses of DNA deformability and correlations between DNA bending and Top6 domain displacement highlight a dynamic interplay between DNA topology and protein conformation in type IIB topoisomerases (Supplementary Tables 3, 4 and Supplementary Fig. 12). These findings collectively suggest that the mechanical state of supercoiled DNA may contribute to the positioning of enzymatic domains responsible for ATP hydrolysis and dimerization and that structural fluctuations in supercoiled DNA can be reciprocally accommodated by dynamic interactions with sites on Top6B. All structures were determined with Top6B in ADPNP-stabilized dimer conformations and exhibited a highly bent DNA state ($> 180°$). However, Top6 is expected to undergo large domain movements during its ATPase cycle, and we anticipate that the dynamic interactions of the enzyme with DNA will accommodate DNA geometries not visualized here. The ability of Top6 to adopt asymmetric forms with DNA may be of utility should the two ATPase sites of Top6 hydrolyze nucleotide asynchronously, a property reported for three other GHKL ATPases: yeast topoisomerase II, bacterial gyrase, and human and zebrafish Hsp90[57–59].

In closing, it should be noted that cleavage of a supercoiled substrate by Top6 requires the enzyme to hold torsional energy constrained in the DNA, as it prevents rotation of the liberated ends. This energy may drive the DNA to bend more sharply at more deformable base pair steps such as those around the cleavage site and, in turn, exert force on the surrounding protein. The response of Top6 to this torsional restraint may be shunted into the Top6B ATPase module, which forms a major interface between the DNA-bound enzyme halves and is seen to twist and shift outward in the most bent conformations. Our reconstructions show that correlated DNA and protein domain movements are accommodated by the transducer helix, which serves as a flexible mechanical element capable of buffering structural strain (Fig. 4b and Supplementary Fig. 11b, c). These findings align with prior suggestions that the transducer and stalk helix help integrate ATPase status with DNA binding and cleavage while also serving as a lever-arm during DNA gate opening[24,30]. The H2TH and GHKL domains also appear to act as flexible, responsive components that shift to accommodate changes in DNA topology, whereas the TOPRIM folds may function as stabilizing elements that constrain motion during catalysis. These relationships suggest that DNA is not a passive participant in the topoisomerase cycle but is instead an active modulator of Top6 architecture, communicating substrate topology to the enzyme. Indeed, DNA torsional strain may influence A-subunit domain movements to promote opening of the DNA gate, a state that was not captured in the current cryoEM data. Energetic linkages between Top6 and DNA may be of value in physiological contexts where supercoiling density fluctuates, such as during replication or transcription. This interpretation comports with the growing understanding that DNA-processing enzymes both harness and exploit the mechanical properties of nucleic acids to drive conformational transitions essential for their catalytic function.

## Methods
### Cloning
A pST39 polycistronic expression vector carrying *Methanosarcina mazei* Top6A and Top6B genes[15,60] was modified using Gibson assembly[61] to replace the 6xHis-TEV-cleavable tag on the N-terminus of the *top6B* gene with a 14xHis-SUMO tag. The Top6$^{(A:E342Q)}$ variant and Top6B stalk-residue variants were made by Around-the-Horn site-directed mutagenesis[62].

## Protein expression and purification

Expression and purification of WT Top6 and all variants followed the same protocol. Top6 was overexpressed in *E. coli* BL21-CodonPlus (DE3)-RIL cells (Agilent) using autoinduction[63]. Cells were grown at 32 °C in Studier ZYM-5052 media in the presence of 0.1 mg/ml kanamycin for 24 hr before harvesting by centrifugation. Cells were resuspended in lysis buffer (20 mM HEPES pH 7.5, 800 mM NaCl, 20 mM imidazole, 10% glycerol, 1 μg/mL leupeptin, 1 μg/mL pepstatin, 1 mM phenylmethylsulfonyl fluoride (PMSF), 1 mg/ml lysozyme, 0.1 mg/ml DNase I) and lysed by sonication. Lysate was clarified by centrifugation at $34,000 \times g$ for 20 min at 4 °C. The supernatant was then filtered through a 1.1 μm glass fiber filter (Thermo Scientific) and loaded onto a 1 mL HisTrap HP column (Cytiva). The column was washed with 20 mL of lysis buffer (without lysozyme and DNase I) and 5 mL of equilibration buffer (20 mM HEPES pH 7.5, 150 mM NaCl, 20 mM imidazole, 10% glycerol, 1 μg/mL leupeptin, 1 μg/mL pepstatin, 1 mM PMSF), followed by a 15 mL gradient elution to 100% elution buffer (20 mM HEPES pH 7.5, 150 mM NaCl, 300 mM imidazole, 10% glycerol, 1 μg/mL leupeptin, 1 μg/mL pepstatin, 1 mM PMSF). Fractions with eluted protein were pooled and applied to tandemly coupled 1 mL HiTrap SP and Q columns (Cytiva). The linked columns were washed with 5 mL of equilibration buffer, then the SP column was detached, and protein was eluted from the Q column with a 10 mL gradient to lysis buffer. The eluted protein was cleaved with His-tagged SENP1 protease (purified in house, final concentration ~ 0.1 mg/mL) for 1 h at 4 °C and concentrated in an Amicon Ultra centrifugal filter with a molecular weight cutoff of 10 kDa. Cleaved and concentrated protein was run over a Superose 6 Increase 10/300 column (Cytiva) equilibrated in size exclusion chromatography buffer (20 mM HEPES pH 7.5, 300 mM KCl, 0.5 mM tris(2-carboxyethyl)phosphine (TCEP), 10% glycerol). Peak fractions were collected and concentrated; when stored frozen, the final glycerol concentration was increased to 30% before flash-freezing in liquid nitrogen and moving to − 80 °C. The molar concentration of Top6 is calculated based on the molecular weight of a $Top6A_2\text{-}Top6B_2$ complex ($\varepsilon = 188010$ A/cm/M for the 220 kDa heterotetramer).

## DNA substrate purification

Two different minicircles were used for screening based on availability; 327 bp minicircle was used for the WT complex and a 306 bp minicircle was used for the Top6[(A:E342Q)] complex. Minicircles were generated in *E. coli* ZYCY10P3S2T cells (System Biosciences) transformed with a parental plasmid containing 32 I-SceI sites and a minicircle sequence flanked by PhiC321 *attB-attP* sites[64], following the growth and induction procedure described therein, and purified using an in-house modified procedure described below. The superhelical density of the mcDNA substrate is the same as the parental plasmid it is derived from ($\sigma = - 0.06$ for *E. coli*), and is expected to give rise to a specific linking difference ($\Delta Lk$) of either − 2 or − 1 ($\Delta Lk = \sigma \times Lk_O = 0.06 \times 327$ bp / 10.5 bp per helical period = 1.87, or $0.06 \times 306$ bp / 10.5 bp per helical period = 1.75). In the gel shown in Supplementary Fig. 1b, the thick supercoiled band likely contains a mixture of − 1 and − 2 species, consistent with findings in Figs. 2 and 3 of Fogg et al.[28]. The persistence of a supercoiled band at high Top6 concentration likely results from a − 1 to + 1 relaxation event, as seen in Supplementary Fig. 2 of Irobalieva et al.[29]. Non-labeled bands in our gels likely correspond to small amounts of minicircle dimers (which can occur in homologous recombination-proficient *E. coli*), residual unrecombined parental plasmid, and genomic DNA that carried over during purification.

Cells were resuspended in 50 mM Tris-HCl, pH 8.0, 10 mM EDTA, 50 mM glucose, 1 mg/ml lysozyme, 0.1 mg/ml RNase A and lysed by mixing with 200 mM NaOH, 1% SDS. Lysis was quenched by mixing with 4 M KOAc, and the lysate was clarified by centrifugation at $15,000 \times g$ for 1 h at 4 °C. The supernatant was filtered through a cloth filter (EMD Millipore Miracloth 4758555-1 R) and precipitated with isopropanol (70% final concentration, incubated at 4 °C overnight). Nucleic acids were pelleted by centrifugation and resuspended in 280 mL of TE buffer (10 mM Tris-HCl, pH 8.0, 1 mM EDTA). Insoluble RNA was removed by an additional centrifugation, and RNA remaining in the supernatant was precipitated by adding 50 mL of 5 M $CaCl_2$ (to a final concentration of 750 mM), incubating at room temperature (RT) for 30 min, and centrifuging at $12,000 \times g$ for 40 min at 4 °C. DNA was precipitated with isopropanol (70% final concentration, incubating at 4 °C for 1 h) before centrifuging at $15,000 \times g$ for 1 hr. The pellet containing DNA was resuspended in 100 mL of DEAE-A buffer (50 mM Tris-HCl, 1 mM EDTA, 250 mM NaCl, pH 7.2). Residual insoluble RNA was removed by centrifugation. Residual soluble RNA was precipitated by $CaCl_2$ (final concentration 75 mM), incubating at RT for 30 min, and centrifuging at $18,000 \times g$ for 30 min.

Following the final RNA removal step, the supernatant was loaded onto a 4 mL CIMmultus DEAE column (Sartorius BIA-414.5114-1.3) equilibrated in DEAE-A buffer. The column was next washed with DEAE-A buffer before step elution by DEAE-B buffer (50 mM Tris-HCl, 1 mM EDTA, 1 M NaCl, pH 7.2). Several repetitions of DEAE column loading and elution were performed, followed by gel analysis of fractions to confirm separation and removal of all RNA.

To remove parental plasmid and genomic DNA from the DNA solution eluted from the DEAE column, NaCl was added to 5 M, $MgCl_2$ was added to 10 mM, and polyethylene glycol (PEG) molecular weight 3350 was added to 6.5% and incubated on ice for 30 min. The desired DNA was pelleted at $2934 \times g$ in a tabletop Sorvall swinging bucket centrifuge for 30 min, after which the pellet was air-dried and resuspended in 1 mL of TE. One mL of phenol chloroform (pH 8.0) was added, and the mixture was vortexed to emulsify it, then centrifuged at $2934 \times g$ for 30 min, leaving the PEG at the air-water interface. The aqueous layer containing DNA was aspirated by pipette. This phenol-chloroform extraction procedure was repeated one more time. Final removal of contaminants was achieved by ethanol precipitation. Minicircle DNA was resuspended in water and placed at 4 °C for storage.

## Plasmid supercoil relaxation

Top6 was thawed and diluted in series with enzyme dilution buffer (20 mM HEPES pH 7.5, 250 mM potassium glutamate, 1 mM TCEP, 10% [v/v] glycerol) and incubated with negatively supercoiled pSG483 plasmid DNA for 5 min on ice. Addition of 2.5 x buffer (40 mM HEPES pH 7.5, 25 mM $MgCl_2$, 2.5 mM TCEP, 0.25 mg/ml BSA, 20% glycerol) and 5x ATP (5 mM ATP) made the final relaxation assay condition of 40 μL reaction volume; 0, 0.05, 0.15, 0.4, 1.3, 4, and 12 nM Top6; 5 nM pSG483 (14.6 μM in terms of base pairs); 20 mM HEPES pH 7.5; 50 mM potassium glutamate; 1 mM $MgCl_2$; 1 mM TCEP; 10% glycerol; 0.1 mg/mL BSA; and 1 mM ATP. After addition of ATP, reactions were incubated at 37 °C for 30 min, then quenched by addition of SDS and EDTA to final concentrations of 1% and 10 mM, respectively, and the protein was digested by 0.3 mg/ml proteinase K at 55 °C for 45 min. To visualize results, samples were mixed with loading dye containing glycerol and then run on a 1.6% (w/v) TAE agarose gel (50 mM Tris-acetate pH 7.9, 40 mM sodium acetate, 1 mM EDTA pH 8.0) for 22 hr at ~ 1.2 V/cm. Gels were stained for 30 min with 0.5 μg/mL ethidium bromide in running buffer, de-stained in running buffer twice for 45 min, and photographed by UV trans-illumination in a Bio-Rad GelDoc Go imager. Experiments were repeated at least twice using separate aliquots of frozen enzyme and freshly prepared reagent mixtures.

## Minicircle supercoil relaxation

The procedure with minicircles was like the procedure with plasmids, except that the Top6 titration in the final relaxation assay condition was 0.08, 0.25, 0.74, 2.2, 6.7, and 20 nM Top6, and the DNA concentration was 20 nM 306 bp minicircle (6.1 μM in terms of base pairs) and a native PAGE technique was used for visualization instead of an

agarose gel. Linearized, and nicked mcDNA controls were made with EcoRV-HF and Nt.BstNBI (New England Biolabs), respectively, in NEB-uffer 3.1. T5 exonuclease (New England Biolabs) in NEBuffer 4 was used to check for the presence of linear or ssDNA species in the mcDNA samples. Samples were mixed with loading dye containing glycerol and then run on a 5% (w/v) acrylamide:bis-acrylamide (29:1) gel with calcium (50 mM Tris-acetate pH 7.9, 40 mM sodium acetate, 10 mM CaCl$_2$) for 2.5 hr at ~ 7.5 V/cm. Gels were stained for 15 min with SYBR Gold in running buffer and photographed by UV trans-illumination in a Bio-Rad GelDoc Go imager. Experiments were repeated at least twice using separate aliquots of frozen enzyme and freshly prepared reagent mixtures.

### Minicircle cleavage

The procedure for cleavage of minicircles was like that of minicircle supercoil relaxation, except that the concentration of Top6 was held at 20 nM and ADPNP was used instead of ATP. Etoposide, which had been reported to inhibit unknotting by *S. shibatae* Top6[9], was also tested in this assay but found to have a relatively negligible effect. For reactions with (or without) etoposide, 0.25 mM etoposide (or water) was added after the 2.5 x reaction buffer and incubated at 37 °C for 30 min, before addition of 1 mM ADPNP. Experiments were repeated at least twice using separate aliquots of frozen enzyme and freshly prepared reagent mixtures.

### CryoEM sample preparation

Top6 did not form stable, complete particles in the open holes of carbon or gold grids. After screening crosslinkers, detergents, various carbon supports, Top6 was found to only form viable particles on amorphous carbon surfaces, with the highest particle quality observed on carbon over holey gold foil. Batches of Au-Flat 1.2/1.3 µm 300-mesh gold-palladium grids (Protochips/Electron Microscopy Sciences) were prepared with a thin amorphous carbon layer as follows: carbon was first deposited onto mica sheets using a Denton DV-502A evaporator, then the thin layer was transferred to the surface of water, and finally lowered onto the surface of the grids, which were left to dry on a sheet of blot paper.

Freshly expressed and purified wildtype Top6 was concentrated to 60 µM (13 mg/ml) and stored on ice during transport by rail to the National Center for CryoEM Access and Training at New York Structural Biology Center. On site, it was diluted in series with enzyme dilution buffer (20 mM HEPES pH 7.5, 250 mM potassium glutamate, 5% [v/v] glycerol) to 0.8 µM in 8 µL and combined with 8 µL of 1.25 µM negatively supercoiled, 327 bp minicircle DNA for 5 min on ice. Then 16 µL of 2.5x buffer (40 mM HEPES pH 7.5, 25 mM MgCl$_2$, 2.5 mM TCEP, 0.25 mg/ml BSA, 20% glycerol) and 5x ADPNP (5 mM) were added, and the mixture was incubated for 5 min at 20 °C after each addition. The final pre-grid condition was a 40 µL volume containing Top6-minicircle at ~184 nM (0.05 mg/ml), 20 mM HEPES pH 7.5, 50 mM potassium glutamate, 1 mM Mg$^{2+}$, 1 mM TCEP, 1% glycerol, 1 mM ADPNP. Minutes before vitrification, the thin-carbon-coated grids were gently plasma-treated in a Gatan Plasma System for 7 s using an ArO$_2$ atmosphere and RF power setting of 15 W. Three µl of sample were applied to the grids, held for 20 s, and blotted for 3–4 s at force 0, corresponding to the pad shadows touching, before plunging into liquid ethane in a Vitrobot Mark IV held at 10 °C and 100% relative humidity.

Grids with Top6$^{(A:E342Q)}$ and negatively supercoiled, 306 bp minicircle were prepared at Johns Hopkins from frozen Top6$^{(A:E342Q)}$ stored at 3.4 µM (0.75 mg/ml). Assembly of the complex and grid preparation followed a similar procedure as the wildtype, except for these differences: The 2.5 x buffer contained 25 mM CaCl$_2$ instead of MgCl$_2$. After adding 2.5 x buffer, the sample was also incubated with etoposide, an agent that promotes DNA cleavage by Top2, at a final concentration of 250 µM for 30 min at 37 °C. Although little cleavage stimulation was

seen for etoposide (Supplementary Fig. 7b), it was nonetheless included as a factor that might help trap a cleavage complex. Following incubation with etoposide, ADPNP was added for 30 minutes at 37 °C. The final concentration of the complex was 200 nM. Grids were gently plasma treated in a Pelco easiGlow for 15 s using air and a current setting of 15 mA before application of 3 µl samples, 20 s adsorption hold time, blotting for 1–4 s at force 8, corresponding to the pad shadows touching, and plunge freezing.

### CryoEM data collection

The wildtype datasets were collected via an NCCAT Rapid Access Proposal with follow-on sessions between January and July 2023 using the same 300 kV Titan Krios with Falcon 4 direct electron detector in counting mode, Selectris energy filter with 10–20 eV slit width, and Leginon 3.6 software, with the same acquisition parameters, exhausting several identically prepared grids. In total, NCCAT staff collected 50,906 EER-format exposures at 0.96 Å/pixel (130kx nominal magnification) and 50 e⁻/Å$^2$ dose over seven seconds, with a nominal defocus range of – 0.8 to – 2.5 µm. Approximately 80% of the wildtype dataset was collected at a 30° stage tilt to attempt to compensate for the preferred orientation of the particle.

The Top6$^{(A:E342Q)}$ dataset was collected at the Johns Hopkins Beckman CryoEM Center in March 2024, on a 300 kV Titan Krios G3i with Falcon 4i direct electron detector in counting mode, Selectris energy filter with 10 eV slit width, and EPU 3.6.0.6389 software. In total, we collected 26,728 EER-format exposures at 0.93 Å/pixel (130kx nominal magnification) and 40 e⁻/Å$^2$ dose over four seconds, with a nominal defocus range of – 1.0 to – 2.0 µm.

### CryoEM data processing

All processing was performed in CryoSPARC v4.4.1 or v4.6.0[65–67]. Workflows for the wildtype and Top6$^{(A:E342Q)}$ datasets shared the following scheme and parameters: We imported movies with fractionation of EER files into 40 frames and EER upsampling factor 1 (none). We performed patch motion correction, patch CTF estimation, and gentle exposure curation, excluding outliers in several CTF parameters. We employed crYOLO (versions 1.9.1–7) to pick particles using the pre-trained model for low-pass filtered cryo images[68]. We extracted particle images in 360-pixel boxes downsampled to 180 pixels. Several iterations of 2D classification led to a large, broad subset to preserve rare orientations and a small subset of high-quality particles, from which 3D ab initio jobs of 2–6 classes revealed symmetric and lopsided maps. We set several custom parameters in 2D classification jobs to increase the persistence and visibility of rare orientations of our somewhat small particle, though it is important to note that we took care not to eliminate particles too aggressively in 2D classification and instead relied on decoy classification in 3D for more thorough cleanup. The custom parameters were generally 100–200 classes, 2 final full iterations, 40 online-EM iterations, batch size per class 400.

Ab initio maps, including several junk maps, were used as references to begin a series of 3D heterogeneous refinement iterations that cleaned up and classified the large particle stack. Each iteration propagated maps from all classes but particles from the symmetric and lopsided classes only. After appreciable cleanup, the particles were re-extracted in 360-pixel boxes and subjected to additional decoy classification. The symmetric and lopsided states were then treated as separate particle stacks and refined in 3D homogeneous and non-uniform refinement and further examined or refined in 3D classification, 3D variability analysis, or local refinement.

For the wildtype dataset, refinement of the lopsided state ended with homogeneous refinement, non-uniform refinement with increased dynamic-mask-far parameter (21 Å), and non-uniform refinement that also optimized over per-group CTF parameters and minimized over per-particle scale. Refinement of the symmetric particle state also proceeded through homogeneous and non-uniform

refinements, but 3D variability analysis revealed, and 3D classification separated, the unfolded-transducer (tilt-symmetric) state. Refinement of the symmetric and tilt-symmetric substates ended with local refinements of globally masked, signal-subtracted particle stacks for each substate.

For the Top6$^{(A:E342Q)}$ dataset, refinement of the lopsided state ended with homogeneous refinement and non-uniform refinement that minimized over per-particle scale. Refinement of the symmetric (cleavage) state also followed those steps but ended after global CTF refinement with fitting of spherical aberration, tetrafoil, and aniso-tropic magnification in two iterations; a non-uniform refinement that minimized over per-particle scale; and a globally masked local refine-ment with Gaussian priors, shift/rotation recentering, and an initial lowpass resolution of 5 Å (following the non-uniform refinement that had already reached 2.5 Å).

Final refinements of all five states were also analyzed by orienta-tion diagnostic[69,70], local resolution[71], and local filtering jobs. For the symmetric states, we opted not to apply symmetry refinement to attempt to achieve higher resolutions at the expense of observing potentially functionally important characteristics.

### Model building and figure generation

We used ModelAngelo v1.0.12[72] to build the Top6$^{(A:E342Q)}$ cleavage state into the CryoSPARC-sharpened map, but merged the C-terminal domains of the B-subunit from Corbett et al.[15], and manually added ADPNP, Ca$^{2+}$, K$^+$, and a DNA oligomer model generated by Coot v0.9.8.92[73] ("Build Nucleic Acid" under "Other Modeling Tools") and moved into place using ChimeraX v1.9[74] and ISOLDE v1.9[75] with adap-tive distance restraints. We corrected missing residues of the Mod-elAngelo model using Coot and corrected geometry errors with ISOLDE and Coot Refine Zone and Angles. For the phosphotyrosine linkage, we used Coot to add a Link record between the tyrosine hydroxyl oxygen and 5′ phosphate (which already contained only 3 oxygens in default nucleic acid models), and we separated the DNA oligomer strands at the broken backbone site by renaming chains. We manually deleted unnecessary hydrogens and used refinement.geo-metry_restraints.edits parameterization during refinement in Phenix v1.21.1-5286[76] to apply the appropriate distance and angle restraints in the phosphotyrosine. We refined in Phenix with 1-2 macrocycles, including global minimization and atomic displacement parameters.

The specific DNA sequence in the cleavage state was determined by first segmenting individual base pairs and nucleotides from the full cryoEM density map. Using an ad hoc method[45] based on structural templates, shape descriptors and size, each individual nucleotide density was categorized as purine (R), pyrimidine (Y), or undetermined (X). Individual nucleotides were then matched to their respective base pair to ensure each base pair contained one purine and one pyr-imidine; base pairs that did not follow standard base-pairing rules were not included in the subsequent analysis. This allowed for a 3D R/Y profile to be generated that could then be compared to the R/Y profile created from the known mcDNA sequence. From this R/Y profile alignment, the exact DNA sequence could be extrapolated.

We then examined mcDNA sequence-dependent deformability values for base pair steps in a tetrameric context[47] using a custom Python program, described in the accompanying paper[45]. To deter-mine average deformability scores of a given sequence, a *k-mer* sliding window tool was used, which enabled comparison among different sequences in the minicircle of given length *k*. For this analysis, *k*-mers of length 30 were chosen, corresponding to the length of the region of the mcDNA bound to the Top6A dimer.

We adapted the cleaved model to uncleaved states using Chi-meraX functions splits, fit-in-map, and combine; ISOLDE with and without adaptive distance restraints, DNA base pairing restraints, and protein secondary structure restraints; and Coot Refine Zone. Adapt-ing models to the lopsided states required extra care with restraints in ISOLDE. During Phenix refinement, to reduce clash scores in models for the lower-quality maps, we set map weight 0.6 instead of the default 0.8–1, and nonbonded weight 2000 instead of the default 100.

Figures were generated with ChimeraX v1.9 and Apple Keynote v14.1. Except for CryoSPARC, we used SBGrid to manage electron microscopy-related, modeling, and graphics software[77].

### DNA bending and domain rearrangement correlation analysis

All five cryoEM-derived models were aligned on the DNA duplex using UCSF ChimeraX. Structural analysis focused on six biochemically defined protein domains: the Top6A winged-helix domain (WHD; chain A/C, residues 1–145), TOPRIM domain (chain A/C, residues 146–369), and four Top6B domains comprising the GHKL ATPase module (chain B/D, residues 1–234), H2TH (helix-two-turn-helix) motif (chain B/D, residues 235–311), transducer domain (chain B/D, residues 312–510), and C-terminal domain (CTD; chain B/D, residues 511–end). For each domain, the center of mass (COM) was calculated from Cα coordinates, and principal component analysis (PCA) was applied to define the dominant orientation axis. Domain displacements and rotations were computed relative to the cleaved Top6$^{(A:E342Q)}$ reference model by comparing COM positions and PCA-derived axes.

Translational displacements and rotational differences for each region were determined relative to the Top6$^{(A:E342Q)}$ cleaved state. COM shifts were computed as Euclidean distances, while inter-model angular differences between region axes were derived from vector dot products.

Local DNA bending was quantified by identifying a common inflection point across all structures, corresponding to the cleaved or kinked region near residues 35–36 on DNA strands F and G and 37–38 on strands H and I. Phosphorus atom coordinates were used to extract two flanking segments of 7 base pairs on either side of this inflection. A principal axis was fit to each segment, and the DNA bend angle was calculated as the angle between the two resulting vectors. This geo-metric definition enabled consistent comparison of DNA curvature across structurally distinct states.

### Reporting summary

Further information on research design is available in the Nature Portfolio Reporting Summary linked to this article.

## Data availability

The structural models and cryoEM density maps generated in this study have been deposited in the PDB and EMDB repositories, respectively, under the following accession codes. Wildtype symmetric state: 9O8P, EMD-70232. Wildtype tilt-symmetric state: 9O8Z, EMD-70239. Wildtype lopsided state: 9O9M, EMD-70259. Top6$^{(A:E342Q)}$ lop-sided state: 9O7O, EMD-70206. Top6$^{(A:E342Q)}$ cleavage state: 9OOG, EMDB EMD-49972. The cryoEM image datasets from which the maps can be derived have been deposited in the EMPIAR repository. Wild-type dataset: EMPIAR-12904. Top6$^{(A:E342Q)}$ dataset: EMPIAR-12914. Source data are provided in this paper.

## Code availability

A custom Python program to analyze sequence-dependent deform-ability values for base pair steps is provided in the accompanying paper by Baker et al.[45].

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

## Acknowledgements

D.E.R. was supported by a National Institutes of Health Kirschstein-NRSA Postdoctoral Fellowship F32 GM128269. T.J.W. was supported by a National Science Foundation graduate research fellowship, DGE 1106400. H.R.J. was supported by a training fellowship from the Houston Area Molecular Biophysics Program (National Institutes of Health grant T32 GM150582 through the Gulf Coast Consortia). M.L.B. was supported by a grant from the National Institutes of Health, P01 GM063210. H.R.J., R.A.E., J.M.F., and L.Z. were supported by National Institutes of Health grant R35 GM141793 (to L.Z.). D.E.R., F.R., C.B., Q.Y., and J.M.B. were supported by National Cancer Institute grant R35 CA263778 (to J.M.B.). CryoEM data for this work were collected at the National Center for CryoEM Access and Training (NCCAT) and the Simons Electron Microscopy Center located at the New York Structural Biology Center (supported by the NIH Common Fund Transformative High Resolution Cryo-Electron Microscopy program (U24 GM129539, and NIGMS R24 GM154192) and by grants from the Simons Foundation (SF349247) and NY State Assembly) and at the Johns Hopkins Beckman Center for Cryo-EM (supported by the Arnold and Mabel Beckman Foundation, the Howard Hughes Medical Institute, the Johns Hopkins University School of Medicine, and private, anonymous donors); the authors are grateful to the staff at both centers. We also thank the staff at the Johns Hopkins School of Medicine Microscopy Facility and the CSIC Center for Biological Research (Madrid) for training in carbon evaporation and CryoSPARC and the SBGrid Consortium for software support.

## Author contributions

D.E.R. performed all biochemical and structural experiments. F.R. and C.B. purified DNA minicircles. Q.Y. performed molecular cloning. H.R.J. and M.L.B. determined the DNA sequence from EM maps. R.A.E. and L.Z. analyzed DNA deformability. T.J.W. designed the E342Q mutant. D.E.R., J.M.F., L.Z., and J.M.B. conceived the project. M.L.B., L.Z., and J.M.B. supervised the work. D.E.R., H.R.J., R.A.E., and M.L.B. drafted the paper, and all authors edited the manuscript.

## Competing interests

J.M.F. and L.Z. are co-inventors on several patents covering minicircle technology and are shareholders in Twister Biotech, Inc. (rebranded as Velvet Therapeutics, Inc.). The authors declare no other competing interests.
