## [Transparent Peer Review file · Nature Communications]

Supercoiled DNA recognition and cleavage control in topoisomerase VI

Corresponding Author: Professor James Berger

Version 0:

Reviewer comments:

Reviewer #1

(Remarks to the Author)

In this manuscript, the Berger lab and colleagues present structures of a type VI topoisomerase, Top6 from the archaeum *Methanosarcina mazei*, bound to DNA. Topoisomerases alter DNA topology by strand passage, with TopoVI enzymes preferentially relaxing negatively supercoiled DNA and decatenating DNA. How TopoVI recognizes its DNA substrates (for example, to preferentially increase DNA helical winding) is not well understood.

Here, the authors report high-resolution structures of Top6A–Top6B heterotetramers bound to minicircle DNA, revealing pronounced DNA deformations reminiscent of those at the tips of plectonemes in supercoiled DNA. These features likely underlie substrate specificity. A mutant complex trapped in the cleavage state shows even sharper DNA kinking and a 2-bp staggered cleavage site. Strikingly, the authors could pinpoint the DNA cleavage position on the minicircle by deducing the pyrimidine/purine sequence pattern, indicating a strong preference for certain cleavage sites—likely reflecting increased local DNA deformability. Together, the data support a model in which DNA deformation and intrinsic deformability govern cleavage by Top6.

The authors also describe the relevant conformational changes within the enzyme subunits/DNA interactions, and provide insightful comparisons to related molecular machines. The study is timely and provides important mechanistic insights into Top6 activation by DNA deformation. It further highlights striking parallels with DNA defense complexes that use TOPRIM domains for DNA cleavage, suggesting that the meiotic DSB inducer Spo11 may function in an analogous manner. The cryo-EM and biochemical data are of high quality, the figures are clearly presented, and the manuscript is well written. Only a few minor points remain to be addressed. Congratulations to the authors on a beautiful and important study!

Specific comments

- Since the authors propose that DNA deformability is important for Top6 cleavage, could this proposition be further supported for example, by generating minicircle DNA substrates with altered bendability and altered preferred cleavage sites?
- For non-specialist readers, it would be helpful to include a schematic illustrating the overall reaction cycle (DNA cutting–strand passage–religation), highlighting the position of gate and transfer DNAs. This could be placed in either the main text or the supplementary material.
- The introduction could provide more context on the key differences between topoisomerase classes and the expected physiological role of Top6.
- Please indicate the electron density corresponding to the ions described in Figure 1.
- Can the authors comment on or test the DNA-binding ability of the different DNA-interaction mutants analyzed in this study?
- The addition of a few schematics could help the readers, particularly in the notions discussed in Figure 5.
- Please show explicitly in the supplementary figures/legends the PDB/EMD accession codes of the final maps that were obtained.
- “Mks/Wadjet” (e.g. l. 28, 60) should be labelled “MksBEFG/Wadjet” to avoid confusion with the housekeeping bacterial condensin MksBEF.
- Please elaborate on the function of the Top6 C-terminus described l. 135-137.
- Figure 1: it would be helpful to see where is the ‘strap’ mentioned l. 122-123.
- Figure S2: can the authors show the mask used for the local refinement?
- Figure 2: please improve the labeling of the figure to facilitate the readability. Moreover, the grey dotted lines in the figures (for example in Fig. 2 f) are particularly difficult to see.

- There is an extra space before the “.” l. 355
- There are some typos in the methods regarding the “μ” (e.g. l. 593, 610, 619).
- What was the blot force used during grid making (l.597-599)?
- L. 632: were both datasets imported with 40 EER frames?
- Supplementary Table 1: please include the CaBLAM, Cβ outliers and Q-scores.
- Typo “proteinx” l. 116
- Typo “notE” l. 126

Reviewer #2

(Remarks to the Author)

Topoisomerase VI (Top6) is a type IIB topoisomerase found mainly in archaea and plants, and shares structural and functional homology with Spo11 and several TOPRIM-based bacterial nucleases. While its general architecture and catalytic activity have been characterized, the molecular mechanisms underlying its DNA recognition, cleavage regulation, and communication with the ATPase domain have remained elusive for many years. In this study, the authors use single-particle cryo-EM to obtain multiple conformational states of *Methanosarcina mazei* Top6 bound to negatively supercoiled DNA minicircles. The structures reveal both cleaved and uncleaved states and uncover several Top6-specific regulatory features, including a WH–TOPRIM latch loop, an electrostatic clasp that controls access to the scissile phosphate, and a transducer stalk that acts as a tension-sensing element. A key conceptual advance is the identification of a sequence-dependent DNA deformability motif that guides site-specific cleavage, highlighting the importance of local DNA structural flexibility in modulating enzyme reactivity. These findings provide important mechanistic insights into Top6 function and have broader implications for Spo11 and related nucleases. I believe this is a strong and significant contribution. I'd recommend publication in Nature Communications pending appropriate revisions to address the following comments and concerns.

(1) The cleaved DNA state presented in this study was obtained using the Top6A E342Q mutant, which disrupts a conserved electrostatic clasp and supposedly biases the enzyme toward the cleaved conformation. This approach is well-justified and highlights the notion that DNA cleavage by Top6 is normally under tight regulatory control. However, it remains unclear to what extent Top6 cleavage activity is stimulated *in vivo*, and what physiological cues, such as DNA supercoiling, accessory factors, or ATPase cycle dynamics, might trigger the conformational transitions necessary for activation. The authors are encouraged to expand their discussion on this point, including any evidence (direct or indirect) that supports *in vivo* regulation of cleavage activity, and to speculate on the possible molecular mechanisms that may drive activation under physiological conditions.

(2) One concern is that the DNA minicircles used for cryo-EM analysis may not fully recapitulate the torsional stress and conformational heterogeneity present in genomic or plasmid DNA *in vivo*, particularly due to their small size. Given that type II topoisomerases, including Top6, are known to preferentially bind DNA crossovers or junctions, it would be helpful to clarify whether the observed Top6–DNA complexes formed preferentially at crossover sites or at the plectoneme apex (i.e., the sharply bent tip of the supercoiled DNA loop). The authors may consider analyzing their raw cryo-EM micrographs or particle classes to estimate the proportion of particles bound at crossovers versus apical loop regions. If Top6 binds predominantly at the apex rather than at DNA crossings, this would suggest that the enzyme is recognizing and stabilizing pre-existing DNA curvature, rather than actively inducing DNA deformation to promote cleavage. Such a result would also raise mechanistic questions: if Top6 lacks intrinsic DNA-shaping activity and does not prefer crossovers, how does it catalyze its hallmark functions, namely DNA relaxation and decatenation, which typically require recognition and manipulation of DNA juxtapositions? Some discussion of this point would strengthen the physiological interpretation of the structural findings.

(3) (Optional) Following the previous comment, the authors may consider performing a DNA cleavage assay using a linear DNA duplex that contains the identified cleavage site. Such an experiment could help clarify whether Top6 requires pre-existing DNA curvature or topological tension (as found in supercoiled minicircles), or whether it can catalyze strand scission on a relaxed linear substrate bearing the same sequence. If efficient cleavage is observed on linear DNA, this would suggest that local sequence-dependent deformability is sufficient to promote Top6 activation, even in the absence of global supercoiling. Conversely, if cleavage is significantly reduced or abolished, it would support the idea that Top6 recognizes and exploits the topological stress or curvature present in supercoiled DNA, rather than reshaping DNA conformation on its own. This comparison could offer valuable insights into the degree to which DNA topology contributes to enzyme activation and cleavage site selection.

(4) To catalyze relaxation or decatenation, type II topoisomerases must transiently open their DNA-gate to allow passage of the T-segment. In the case of Top6, the DNA-gate is formed by the Top6A dimer interface, which appears to be both structurally extensive and conformationally stable in the current cryo-EM structures, even in the cleaved state. This raises questions about how DNA-gate opening is initiated and regulated in the context of the full catalytic cycle. The authors are encouraged to discuss potential mechanisms for gate opening based on features observed in their structures. For example, are there signs of conformational strain, hinge-like flexibility, or latent asymmetry that might enable partial destabilization of the Top6A dimer interface under nucleotide or DNA geometry control? Could transient disruption of the WH–TOPRIM or stalk interactions promote gate separation? Additionally, do any of the asymmetric conformations observed in this study suggest a pathway toward DNA-gate opening?

(5) In type IIA topoisomerases, cleavage site selection and catalysis are closely linked to local DNA structural transitions, particularly the adoption of A-like DNA geometry around the scissile site. This conformational change is promoted in part by the intercalation of conserved isoleucine (or valine) residues into the minor groove, which introduces local underwinding and bends the DNA to favor catalysis. In the current study, the authors report that DNA bound by Top6 also undergoes localized deformation toward an A-like geometry near the cleavage site, yet no comparable base pair intercalation is observed in the structures. This raises an intriguing mechanistic question: what structural features in Top6 drive this deformation in the absence of intercalating residues? Is the DNA geometry a consequence of the overall supercoiled context and loop positioning, or are there protein-induced constraints (such as contacts from the latch loop), WH domain positioning, or charge-based groove compression that help promote A-like character? The authors are encouraged to speculate on the molecular origin of this conformational transition, and whether it reflects a fundamentally different mode of cleavage site activation compared to type IIA enzymes.

Minor points:

Page 6, line 126: change "note" to "not"

Page 29, line 619: μ instead of "u"

Reviewer #3

(Remarks to the Author)

Richman and coworkers present single-particle cryo-EM based structures of the prototypical topoisomerase IIB, topo VI, bound to supercoiled minicircle DNA substrates. This work provides important insights into the binding and cleavage of supercoiled DNA by a type IIB topoisomerase. Moreover, the authors have achieved a number of important novel results that likely have broader implications for other type II topoisomerases, and homologous proteins including the meiotic cleavage factor Spo11. Highlights of these new results include the identification of a single amino acid mutation that results in the stabilization of a cleaved gate-segment DNA intermediate of the originally intact mini circle DNA. Through additional class averaging and refining the cryo-EM model building, the authors were able to identify several asymmetric states of the heterotetramer bound to both cleaved and intact gate segment DNA. Finally, the authors employ new (co-submitted) approaches to resolve the type of DNA bases (purine or pyrimidine) resolved in the cryo-em structure, which allowed them to define an apparently unique cleavage site on the mini-circle DNA. Applying another new approach to computationally obtain DNA deformability, they relate the cleavage site observed on the mini circle DNA to a specific pattern of DNA deformability. These results together reveal important new insights into the plasticity and dynamics of topo VI binding to, and cleaving, supercoiled DNA.

Overall the work is exceptionally well-conceived and well-executed. The conclusions largely follow from and are well-supported by the data. The structural insights are novel and some, such as the identification of asymmetric structure and protein plasticity, represent advances in the topoisomerase field more generally.

Despite my overall enthusiasm, there are a number of mostly minor points that I think the authors should address prior to publication. Many of these points focus on the use of the supercoiled mini-circle DNA as the substrate for the Cryo-EM studies. The overarching question and potential caveat associated with using supercoiled minicircles relates to the degree of supercoiling (or supercoiling density) that potentially drives the protein to adopt structures due to the high degree of twist energy stored in the DNA. These concerns do not derail the work, but they should be addressed directly by the authors. The details of these concerns are described in the individual points below but I wanted to provide a larger framing of the general concerns with using minicircle DNA that are not unique to this work but that should nonetheless be addressed.

1. Line 87 -89 : Topo IV has been shown by Wendorf et al (24) and McKie et al (ref 25) to specifically bind and be activated at DNA crossovers, rather than plectoneme ends. These findings are at variance with the statement that Topo IV binding to a loop resembling the end of a plectoneme could account for the preference of the enzyme for supercoiled substrates. A similar argument is forwarded in lines 159-161, "Overall, the degree of DNA bending observed resembles that expected at the apical tip of a plectonemic supercoil, potentially accounting for why Top6 shows a preference for binding supercoiled substrates."

2. Line 105 – methods, and Supplemental figure 1: What is the topological state of the minicircles? Can the apparent distribution of topoisomers apparent in supplementary figure 1 be assigned and the resulting supercoiling density be provided? This is perhaps a feature of the mc DNA substrates, but it is surprising that the supercoiled bands run slower than the linear band. A citation to the relevant literature or a brief explanation of this counterintuitive behavior would be helpful understanding supplementary figure 1. There appear to be doublets for each supercoiled band in the gel in supplementary figure 1, can the authors comment on these doublets? The relaxation activity of the supercoiled mc DNA is unusual in that the intermediate state seems to entirely be relaxed or nicked at high topo VI concentrations, whereas the more extensively supercoiled species is reduced, but remains. Assignment of the topoisomers would be helpful in understanding this relaxation pattern. Some of the EM structures were obtained in the presence of ADPNP. In type IIA topoisomerases, non-hydrolysable ATP analogs can support a single strand passage event observed after protein removal. It would be helpful to provide a similar gel as in supplementary figure 1 in the presence of ADPNP to verify the topological state of the mc DNA in the cryo-EM structures. From Supplementary Figure 7 it appears that AMPPNP indeed supports limited relaxation of the mcDNA substrate, though this point is not elaborated in the manuscript. Finally, it would be helpful if the authors could provide the distribution of mc DNA topoisomers bound to topo VI in the absence and presence of the non-hydrolysable nucleotide. This would help define substrate in the bound structures. Alternatively, or additionally, are there structural indicators of the topological state of the DNA in the obtained structures?

3. How does the relaxation rate of the mc circle DNA compare with plasmid DNA?

4. Line 126 "note" should be "not" .

5. Supplementary Fig. 7. It is interesting that the Top6(A:E342Q) mutant exhibits limited relaxation activity in the presence of AMPPNP along with Mg and to a more limited degree Ca. It would be helpful to provide similar gels in the presence of ADPNP to ascertain the degree of relaxation activity with this co-factor used in the structural determination. It would also be good to establish that the Top6(A:E342Q) mutant supports supercoil relaxation, beyond what can be inferred from the gel in Supplementary Fig. 7.

6. Supplementary table 2. This table requires more explanation: What do the colored bars (yellow and red) represent? What do the horizontal lines represent? For orientation it would be helpful to include the 3' and 5' designations of the chains.

7. 245: "This finding establishes that Top6 associates with a specific region of the mcDNA in the 245 cleavage state (Supplementary Fig. 10b)." Have similar minicircles been imaged previously, or the bending and distortion locations of these minicircles been previously determined? This question is aimed at understanding to the degree to which the structure is imposed by the protein versus how much it is imposed by the highly constrained and possibly sharply bent mcDNA. The location of the binding for example could represent the pre-formed bent configuration of the DNA that the protein can effectively bind as opposed to a sequence and structural preference of the enzyme. This possibility should be discussed and either addressed through perhaps previous data or mentioned as a potential caveat.

8. Supplementary Figure 10 and lines 254-274. The deformability analysis is an interesting and valuable addition to this work. It would be helpful to provide a mechanistic description of deformability to help orient the reader. It would also be helpful to more closely align the conclusions with the figures. For example, it would be helpful to indicate the cleavage site on the deformability plots (the cleavage region at least) and it would be helpful to graphically demonstrate the differences in deformability with respect to the cleavage site on one or more of the plots. As is, it is not entirely clear how to connect the data in the figure with the conclusions based on the data. Finally, there seems to be some confusion with respect to the scales of the deformability plots. The y-axis is labeled as Deformability relative to average. In panel C this is strictly positive, whereas in panel D it takes on positive and negative values. Finally, in the text there is explicit mention of particular deformability values, which are impossible to compare since the plots in the figure report relative values rather than absolute values. Cleaning up the scales and better correlating the regions mentioned in the text with corresponding well-defined regions in the figure will make these results significantly easier to understand and will make the points that the authors are making much clearer. One final analysis that could help increase the statistical significance of the finding, would be to determine the uniqueness of the bendability feature that the authors claim is driving the cleavage and demonstrate that it is unique along the mcDNA.

Version 1:

Reviewer comments:

Reviewer #1

(Remarks to the Author)

The authors have addressed our concerns and further improved the manuscript. We look forward to seeing the paper published.

Reviewer #2

(Remarks to the Author)

All my points were appropriately addressed by the authors. I'd like to recommend acceptance of the revised manuscript for publication.

Reviewer #3

(Remarks to the Author)

The Authors have adequately addressed the comments and requests from the reviewers. I recommend publication of the revised manuscript.

Reviewer #4

(Remarks to the Author)

We would very much like to thank the referees for their time and helpful comments. Responses to each point raised during the review are highlighted below in blue text.

Reviewer #1:

The cryo-EM and biochemical data are of high quality, the figures are clearly presented, and the manuscript is well written. Only a few minor points remain to be addressed. Congratulations to the authors on a beautiful and important study!

We are very grateful to the referee for their kind words and enthusiasm.

- Since the authors propose that DNA deformability is important for Top6 cleavage, could this proposition be further supported for example, by generating minicircle DNA substrates with altered bendability and altered preferred cleavage sites?

This is an interesting and important question. There are many biologically significant sequences that could in principle be examined through the minicircle substrates, such as phased A/T tracts, GC-rich elements, and cleavage sites reported for Top6 by other groups. A systematic analysis of these elements is presently beyond the scope of current work but would make for a very compelling future study.

- For non-specialist readers, it would be helpful to include a schematic illustrating the overall reaction cycle (DNA cutting–strand passage–religation), highlighting the position of gate and transfer DNAs. This could be placed in either the main text or the supplementary material.

Thank you for the suggestion. We agree and have added the requested illustration to Supplemental Fig. 1 of the revised manuscript.

- The introduction could provide more context on the key differences between topoisomerase classes and the expected physiological role of Top6.

We agree - we initially included more material on these topics but ultimately needed to remove it due to length constraints. We have instead directed readers to recent reviews, especially the section on Top6 in McKie *et al.* 2021, and primary source citations.

- Please indicate the electron density corresponding to the ions described in Figure 1.

We have revised Fig. 1d and Fig. 3c, which is the analogous view in the cleavage state, to include EM density.

- Can the authors comment on or test the DNA-binding ability of the different DNA-interaction mutants analyzed in this study?

We agree that it would be interesting to try to discern the extent to which specific protein-DNA interactions revealed by the structures contribute to binding or bending, and we initially considered doing so while drafting the present work. However, we became concerned that the number of assays that would be required to systematically assess the impact of all of the new contacts that we observe for both binding and bending would add considerably to the length of the paper, particularly when these experiments would also benefit from assessing their reciprocal impact on ATPase activity to better understand the linkage between nucleotide turnover and substrate recognition. Consequently, we elected to focus experimentally on whether overall catalytic activity (as defined by supercoil relaxation) depends on newly identified features. It is our hope to conduct the suggested biochemical studies as part of a future set of experiments.

- *The addition of a few schematics could help the readers, particularly in the notions discussed in Figure 5.*

Thank you for the suggestion. We have now rearranged Fig. 5 to be full width and include panel 5e, a schematic showing the relationship between transducer tightness and A-subunit activation with respect to the fold status of the transducer stalk.

- *Please show explicitly in the supplementary figures/legends the PDB/EMD accession codes of the final maps that were obtained.*

This has been added to all relevant main and supplementary figures or legends.

- *“Mks/Wadjet” (e.g. l. 28, 60) should be labelled “MksBEFG/Wadjet” to avoid confusion with the housekeeping bacterial condensin MksBEF.*

Found and fixed at all instances.

- *Please elaborate on the function of the Top6 C-terminus described l. 135-137.*

We have added the phrase ‘*and which may help localize Top6 to a partner protein or subcellular region*’ in that sentence and cited Corbett *et al.* 2007, which elaborates on the question of CTD function.

- *Figure 1: it would be helpful to see where is the ‘strap’ mentioned l. 122-123.*

We have added a ‘top’ view of the B-subunit dimer, featuring the straps, as Supplementary Fig. 4a (and moved the MutL comparison panel to 4b).

- *Figure S2: can the authors show the mask used for the local refinement?*

This is now added. We also added the term ‘global mask’ to that part of the figure to clarify that while we used CryoSPARC’s Local Refinement, which is specialized for tiny (‘local’) angular and translational pose searches, it was used on the global map, not focused on a subregion.

- Figure 2: please improve the labeling of the figure to facilitate the readability. Moreover, the grey dotted lines in the figures (for example in Fig. 2 f) are particularly difficult to see.
- There is an extra space before the "." l. 355
- There are some typos in the methods regarding the " μ " (e.g. l. 593, 610, 619).
- What was the blot force used during grid making (l.597-599)?
- L. 632: were both datasets imported with 40 EER frames?
- Supplementary Table 1: please include the CaBLAM, C β outliers and Q-scores.
- Typo "proteinx" l. 116
- Typo "notE" l. 126

We have checked and fixed all the above. Thank you.

Reviewer #2:

A key conceptual advance is the identification of a sequence-dependent DNA deformability motif that guides site-specific cleavage, highlighting the importance of local DNA structural flexibility in modulating enzyme reactivity. These findings provide important mechanistic insights into Top6 function and have broader implications for Spo11 and related nucleases. I believe this is a strong and significant contribution. I'd recommend publication in Nature Communications pending appropriate revisions to address the following comments and concerns.

We thank the referee for their positive comments and perspective.

(1) The cleaved DNA state presented in this study was obtained using the Top6A E342Q mutant, which disrupts a conserved electrostatic clasp and supposedly biases the enzyme toward the cleaved conformation. This approach is well-justified and highlights the notion that DNA cleavage by Top6 is normally under tight regulatory control. However, it remains unclear to what extent Top6 cleavage activity is stimulated in vivo, and what physiological cues, such as DNA supercoiling, accessory factors, or ATPase cycle dynamics, might trigger the conformational transitions necessary for activation. The authors are encouraged to expand their discussion on this point, including any evidence (direct or indirect) that supports in vivo regulation of cleavage activity, and to speculate on the possible molecular mechanisms that may drive activation under physiological conditions.

Thank you for highlighting this issue. We have expanded the second paragraph in the Discussion (~lines 529-538) to address the points noted.

(2) One concern is that the DNA minicircles used for cryo-EM analysis may not fully recapitulate the torsional stress and conformational heterogeneity present in genomic or plasmid DNA in vivo, particularly due to their small size. Given that type II topoisomerases, including Top6, are known to preferentially bind DNA crossovers or junctions, it would be helpful to clarify whether the observed Top6-DNA complexes formed preferentially at crossover sites or at the plectoneme apex (i.e., the sharply bent tip of the supercoiled DNA loop). The authors may consider analyzing their raw cryo-EM micrographs or particle classes to estimate the proportion of particles bound at crossovers versus apical loop regions. If Top6 binds

predominantly at the apex rather than at DNA crossings, this would suggest that the enzyme is recognizing and stabilizing pre-existing DNA curvature, rather than actively inducing DNA deformation to promote cleavage. Such a result would also raise mechanistic questions: if Top6 lacks intrinsic DNA-shaping activity and does not prefer crossovers, how does it catalyze its hallmark functions, namely DNA relaxation and decatenation, which typically require recognition and manipulation of DNA juxtapositions? Some discussion of this point would strengthen the physiological interpretation of the structural findings.

These are important points. We have now added clarifying language to better address such questions, many of which have been examined in prior biochemical work showing that Top6 prefers binding 1) supercoiled DNA over linear segments, 2) long segments over short, and 3) crossovers over linear segments (Wendorff *et al.* 2018 and McKie *et al.* 2022). These studies also showed that Top6 can promote the bending of a linear DNA segment and that it prefers performing decatenation over supercoil relaxation. Top6 engages the transport segment using specific residues in its central cavity and associates with the bent gate segment using specific residues along the outside of the B-subunit (including ones we now observed in structures). The conclusions of these studies were that Top6 can actively bend DNA, but that it will also preferentially bind to segments that are pre-bent and/or contain a crossover.

In the Results (lines 180-185), we have added clarifying language about our observations, namely that binding to a plectoneme-like loop or a sharp kink of a supercoiled minicircle potentially accounts for the preference for supercoiled substrates over linear or relaxed ones. We also added language in the Discussion (lines 551-560) to reconcile Wendorff's and McKie's observations with our structures, explaining that while Top6 may have a primary preference for catenane crossovers, its secondary preference for supercoils can arise from binding the bent DNA associated with supercoiled substrates, in addition to binding supercoil crossovers.

Unfortunately, we were unable to visualize DNA segments extending beyond the Top6 complex in our raw cryoEM images. In 2D classes, the only DNA we observed was the gate segment 'arms' that bent through and around the sides of the enzyme. We never found evidence for Top6 associating with a crossover in these data sets. It is unclear whether this is because Top6 preferentially localizes to the bends at the apical plectoneme tips of an mcDNA, or whether enzymes that had bound to DNA crossovers removed them during a single cycle of strand passage (supported by ADPNP) before becoming trapped on DNA and bending it. We presently do not have a straightforward set of experiments that could distinguish between these two possibilities.

(3) (Optional) *Following the previous comment, the authors may consider performing a DNA cleavage assay using a linear DNA duplex that contains the identified cleavage site. Such an experiment could help clarify whether Top6 requires pre-existing DNA curvature or topological tension (as found in supercoiled minicircles), or whether it can catalyze strand scission on a relaxed linear substrate bearing the same sequence. If efficient cleavage is observed on linear DNA, this would suggest that local sequence-dependent deformability is sufficient to promote Top6 activation, even in the absence of global supercoiling. Conversely, if cleavage is significantly reduced or abolished, it would support the idea that Top6 recognizes and exploits*

the topological stress or curvature present in supercoiled DNA, rather than reshaping DNA conformation on its own. This comparison could offer valuable insights into the degree to which DNA topology contributes to enzyme activation and cleavage site selection.

This is a good suggestion and dovetails with a similar suggestion from Reviewer 1. As noted above, we aim to perform this study alongside a series of other types of DNA sequences as part of a future set of biochemical experiments.

(4) To catalyze relaxation or decatenation, type II topoisomerases must transiently open their DNA-gate to allow passage of the T-segment. In the case of Top6, the DNA-gate is formed by the Top6A dimer interface, which appears to be both structurally extensive and conformationally stable in the current cryo-EM structures, even in the cleaved state. This raises questions about how DNA-gate opening is initiated and regulated in the context of the full catalytic cycle. The authors are encouraged to discuss potential mechanisms for gate opening based on features observed in their structures. For example, are there signs of conformational strain, hinge-like flexibility, or latent asymmetry that might enable partial destabilization of the Top6A dimer interface under nucleotide or DNA geometry control? Could transient disruption of the WH-TOPRIM or stalk interactions promote gate separation? Additionally, do any of the asymmetric conformations observed in this study suggest a pathway toward DNA-gate opening?

This is an interesting point and one that has intrigued us as well. Unfortunately, we don't yet have clear physical insights as to how the DNA gate opens. We added some speculation on gate opening in the last paragraph of the Discussion in the context of conformational strain and DNA geometry control. We do not have sufficient resolution in the asymmetric states to speculate further about destabilization of domain interfaces, but we find no evidence of DNA separation that might suggest that the system is poised for gate opening. This is something we hope to address in a future study, but it may require some type of specialized DNA substrate or a Top6 mutation that favors the stable formation of such a state so that it may be imaged.

(5) In type IIA topoisomerases, cleavage site selection and catalysis are closely linked to local DNA structural transitions, particularly the adoption of A-like DNA geometry around the scissile site. This conformational change is promoted in part by the intercalation of conserved isoleucine (or valine) residues into the minor groove, which introduces local underwinding and bends the DNA to favor catalysis. In the current study, the authors report that DNA bound by Top6 also undergoes localized deformation toward an A-like geometry near the cleavage site, yet no comparable base pair intercalation is observed in the structures. This raises an intriguing mechanistic question: what structural features in Top6 drive this deformation in the absence of intercalating residues? Is the DNA geometry a consequence of the overall supercoiled context and loop positioning, or are there protein-induced constraints (such as contacts from the latch loop), WH domain positioning, or charge-based groove compression that help promote A-like character? The authors are encouraged to speculate on the molecular origin of this conformational transition, and whether it reflects a fundamentally different mode of cleavage site activation compared to type IIA enzymes.

We apologize for being confusing on this point: We indeed observe that there are loops from the A subunit that partly wedge into and deform the minor grooves near the apex of the bend (Fig.

2c). This action, along with other steric and charge-based interactions along the A-subunit groove (Supplementary Figs. 5c and 10a), help bend DNA but in a markedly different manner from type IIA enzymes (Fig. 6d). As discussed elsewhere in the manuscript and this response, we also speculate that the DNA deformation is intrinsic to the substrate and exploited/stabilized by the protein. We have amended the manuscript to better clarify this point.

Minor points:

Page 6, line 126: change "note" to "not"

Page 29, line 619: μ instead of "u"

We have fixed these and other missing ' μ /u' errors, thank you.

Reviewer #3:

Overall the work is exceptionally well-conceived and well-executed. The conclusions largely follow from and are well-supported by the data. The structural insights are novel and some, such as the identification of asymmetric structure and protein plasticity, represent advances in the topoisomerase field more generally. Despite my overall enthusiasm, there are a number of mostly minor points that I think the authors should address prior to publication. Many of these points focus on the use of the supercoiled mini-circle DNA as the substrate for the Cryo-EM studies. The overarching question and potential caveat associated with using supercoiled minicircles relates to the degree of supercoiling (or supercoiling density) that potentially drives the protein to adopt structures due to the high degree of twist energy stored in the DNA. These concerns do not derail the work, but they should be addressed directly by the authors. The details of the concerns are described in the individual points below but I wanted to provide a larger framing of the general concerns with using minicircle DNA that are not unique to this work but that should nonetheless be addressed.

We understand. We thank the referee for their generally positive view and have endeavored to address specific points noted below.

1. Line 87-89 : *Topo IV has been shown by Wendorff et al (24) and McKie et al (ref 25) to specifically bind and be activated at DNA crossovers, rather than plectoneme ends. These findings are at variance with the statement that Topo IV binding to a loop resembling the end of a plectoneme could account for the preference of the enzyme for supercoiled substrates. A similar argument is forwarded in lines 159-161, "Overall, the degree of DNA bending observed resembles that expected at the apical tip of a plectonemic supercoil, potentially accounting for why Top6 shows a preference for binding supercoiled substrates."*

We apologize for the lack of clarity on this point. It is true that several studies have shown that DNA crossovers are a preferred substrate for binding by Top6. However, Wendorff *et al.* also showed that Top6 can bend DNA segments using the outer contacts along the B subunits that are now observed here. The fact that both bends and crossovers are present in supercoiled DNA has been noted by both Wendorff *et al.* and McKie *et al.* Interestingly, these preferences are paralleled by type IIA topoisomerases, which also bend the DNA targets (G-segments) that they cleave, consistent with the idea that both topoisomerase families evolved structural mechanisms to guide

them to the appropriate topological substrates that require their attention. We have clarified the introduction (lines 63-65) to more clearly state what prior studies revealed about Top6's preference for bends and crossings. This helps frame subsequent statements about plectonemes on lines 87-89 and 162-167, where we discuss how binding to a plectoneme-like loop potentially accounts for the preference for supercoiled substrates in the absence of a crossover.

2. Line 105 – methods, and Supplemental figure 1: What is the topological state of the minicircles? Can the apparent distribution of topoisomers apparent in supplementary figure 1 be assigned and the resulting supercoiling density be provided? This is perhaps a feature of the mc DNA substrates, but it is surprising that the supercoiled bands run slower than the linear band. A citation to the relevant literature or a brief explanation of this counterintuitive behavior would be helpful understanding supplementary figure 1. There appear to be doublets for each supercoiled band in the gel in supplementary figure 1, can the authors comment on these doublets? The relaxation activity of the supercoiled mc DNA is unusual in that the intermediate state seems to entirely be relaxed or nicked at high topo VI concentrations, whereas the more extensively supercoiled species is reduced, but remains. Assignment of the topoisomers would be helpful in understanding this relaxation pattern.

We appreciate these questions. The minicircles have the same supercoil density as the parental plasmid they are released from, the typical *E. coli* supercoil density of -0.06. Because of their small size, 306 bp, they are expected to have a specific linking difference (ΔLk) of approximately -2. The thick supercoiled band likely contains a mixture of -1 and -2 species, consistent with findings from Fogg *et al.* 2006, 'Exploring writhe in supercoiled minicircles' (see Figs. 2 and 3 in that work). The persistence of a supercoiled band at high Top6 concentration likely results from a -1 to +1 relaxation event, as seen in Supplementary Fig. 2 of Irobalieva *et al.* 2015, 'Structural diversity of supercoiled DNA'. Figure 3 of Fogg *et al.* 2006 and Fig. 1 of Irobalieva *et al.* 2015 also show supercoiled species with these smaller linking numbers running slower than the linearized DNA. Non-labeled bands in our gel are likely to correspond to small amounts of minicircle concatemers (i.e., dimers, not catenanes), the parental plasmid, and genomic DNA that carried over during purification. We have amended the Methods section to clarify these aspects of the data.

Some of the EM structures were obtained in the presence of ADPNP. In type IIA topoisomerases, non-hydrolysable ATP analogs can support a single strand passage event observed after protein removal. It would be helpful to provide a similar gel as in supplementary figure 1 in the presence of ADPNP to verify the topological state of the mc DNA in the cryo-EM structures. From Supplementary Figure 7 it appears that AMPPNP indeed supports limited relaxation of the mcDNA substrate, though this point is not elaborated in the manuscript. Finally, it would be helpful if the authors could provide the distribution of mc DNA topoisomers bound to topo VI in the absence and presence of the non-hydrolysable nucleotide. This would help define substrate in the bound structures. Alternatively, or additionally, are there structural indicators of the topological state of the DNA in the obtained structures?

Thank you for raising this question. As pointed out, the cleavage assay of Supplementary Fig. 7 shows that ADPNP supports relaxation of the minicircle for both the WT and E342Q variant. This cleavage assay is also consistent with the relaxation assay of Supplementary Fig. 1 in

demonstrating the topological status of the minicircle, showing the supercoiled, nicked, and relaxed species of the minicircle in the absence and presence of ADPNP and in the absence of enzyme for comparison.

3. *How does the relaxation rate of the mc DNA compare with plasmid DNA?*

This is an interesting question but one that is tricky to address. The supercoil relaxation rate of Top6 is on the order of ~1-6 strand passage events per minute (Wendorff *et al.* 2018, McKie *et al.* 2022). Assessing specific rate differences would require rapid time-resolved studies that we do not currently have in place. While we have thought about establishing such an assay, it is presently beyond the scope of the current work.

4. *Line 126 "note" should be "not".*

Fixed, thank you.

5. *Supplementary Fig. 7. It is interesting that the Top6(A:E342Q) mutant exhibits limited relaxation activity in the presence of AMPPNP along with Mg and to a more limited degree Ca. It would be helpful to provide similar gels in the presence of ADPNP to ascertain the degree of relaxation activity with this co-factor used in the structural determination. It would also be good to establish that the Top6(A:E342Q) mutant supports supercoil relaxation, beyond what can be inferred from the gel in Supplementary Fig. 7.*

We apologize for the confusion in using the terms AMPPNP and ADPNP interchangeably for the same molecule; we have fixed this inconsistency. The minicircle relaxation activity of Top6(A:E342Q) seen in Supplementary Fig. 7 is essentially the same as that of WT under the same conditions.

6. *Supplementary table 2. This table requires more explanation: What do the colored bars (yellow and red) represent? What do the horizontal lines represent? For orientation it would be helpful to include the 3' and 5' designations of the chains.*

Thank you - explanations and annotations have now been added.

7. 245: *"This finding establishes that Top6 associates with a specific region of the mcDNA in the cleavage state (Supplementary Fig. 10b)." Have similar minicircles been imaged previously, or the bending and distortion locations of these minicircles been previously determined? This question is aimed at understanding to the degree to which the structure is imposed by the protein versus how much it is imposed by the highly constrained and possibly sharply bent mcDNA. The location of the binding for example could represent the pre-formed bent configuration of the DNA that the protein can effectively bind as opposed to a sequence and structural preference of the enzyme. This possibility should be discussed and either addressed through perhaps previous data or mentioned as a potential caveat.*

Similar minicircles have been imaged by AFM and show the existence of sharp kinks or plectoneme-like loops (Pyne *et al.* Nature Communications, 2021). Wendorff *et al.* 2018, showed

that Top6 recognizes and exploits pre-existing curvature and also further bends DNA. We have modified the paragraph that follows the quoted statement to reflect these findings and set up the description of the analysis of DNA deformability. We have similarly clarified other parts of the Introduction, Results, and Discussion.

8. *Supplementary Figure 10 and lines 254-274. The deformability analysis is an interesting and valuable addition to this work. It would be helpful to provide a mechanistic description of deformability to help orient the reader.*

We have made the following edits (indicated in red) to the paper in response to this request. We hope that we have captured what the reviewer meant by mechanistic description of deformability.

Results section: ‘DNA sequence is known to significantly impact the ability of DNA to be bent and/or compressed. Using DNA-protein crystal structures, the inherent deformability of specific DNA sequences has been described numerically by the average volume taken up for each unique base pair step. This is based on the idea that base pair steps that are easier to deform into a wider range of shapes take up more volume on average. Sequence-dependent deformability values have been described for individual base pair steps; however, it has been less evident how such values change as a function of DNA segment length. Reasoning that the propensity of a given DNA sequence to bend and deform more readily than others might contribute to the observed binding site preference of Top6, we applied a newly developed computational tool (Baker *et al.*, 2025, accompanying paper) to scan the sequence of the negatively supercoiled minicircle substrate for a pattern of DNA sequence-dependent deformability in the Top6 binding site.’

Methods section: ‘We then examined mcDNA sequence-dependent deformability values for base pair steps in a tetrameric context using a custom Python program, described in Baker *et al.* 2025, accompanying paper. To determine average deformability scores of a given sequence, a *k*-mer sliding window tool was used, which enabled comparison among different sequences in the minicircle of given length *k*. For this analysis, *k*-mers of length 30 were chosen, corresponding to the length of the region of the mcDNA bound to the Top6A dimer...’

It would also be helpful to more closely align the conclusions with the figures. For example, it would be helpful to indicate the cleavage site on the deformability plots (the cleavage region at least) and it would be helpful to graphically demonstrate the differences in deformability with respect to the cleavage site on one or more of the plots. As is, it is not entirely clear how to connect the data in the figure with the conclusions based on the data.

In the cleavage-region inset, we have labeled the steps associated with the cleaved sites so that readers can more readily locate them when reading about their deformability values, 4.6 degrees³•Å³, in the main text.

Finally, there seems to be some confusion with respect to the scales of the deformability plots. The y-axis is labeled as Deformability relative to average. In panel C this is strictly positive, whereas in panel D it takes on positive and negative values. Finally, in the text there is explicit mention of particular deformability values, which are impossible to compare since the plots in the figure report relative values rather than

absolute values. Cleaning up the scales and better correlating the regions mentioned in the text with corresponding well-defined regions in the figure will make these results significantly easier to understand and will make the points that the authors are making much clearer.

We have brought the plots into agreement with a shared Y-axis reference frame and more easily visible average deformability line. Note in comparing panel d to c that the range of the Y axis is smaller (there is less dynamic range in the average value) when the sliding averaging window is larger in panel d.

One final analysis that could help increase the statistical significance of the finding, would be to determine the uniqueness of the bendability feature that the authors claim is driving the cleavage and demonstrate that it is unique along the mcDNA.

We did note that there is one other region of the mcDNA that has a similar pattern and which is located on the opposite side of the mcDNA from the cleavage site, and we labeled both regions in Supplementary Fig. 10c-d. We have edited the Results section to be both more specific and circumspect: 'Interestingly, there is only one other region of the mcDNA that has a similar pattern, which is located on the opposite side of the mcDNA from the cleavage site (Supplementary Fig. 10c), but the central tetramer in this segment lacks the symmetric above-average deformability exhibited by the cleavage site. Thus, while our current model of deformability is unlikely to capture all of the biophysical properties of the mcDNA, it nonetheless indicates that Top6 appears to preferentially bind and cleave at the intersection between highly deformable and relatively rigid DNA segments, particularly when this intersection is itself deformable.'